# Robo2VLM:
# Improving Visual Question Answering using Large-Scale Robot Manipulation Data

**Kaiyuan Chen**[1,*]    **Shuangyu Xie**[1,*]    **Zehan Ma**[1]    **Pannag R Sanketi**[2]    **Ken Goldberg**[1]

[1]University of California, Berkeley    [2]Google DeepMind    *Equal contribution

{kych, syxie, zehanma, goldberg}@berkeley.edu    psanketi@google.com

https://berkeleyautomation.github.io/robo2vlm/

## Abstract

Vision-Language Models (VLMs) acquire real-world knowledge and general reasoning ability through Internet-scale image-text corpora. They can augment robotic systems with scene understanding and task planning, and assist visuomotor policies that are trained on robot trajectory data. We explore the reverse paradigm — using rich, real, multi-modal robot trajectory data to enhance and evaluate VLMs. In this paper, we present Robo2VLM, a Visual Question Answering (VQA) dataset generation framework for VLMs. Given a human tele-operated robot demonstration with video and robot data, Robo2VLM derives ground-truth from non-visual and non-descriptive sensory modalities, such as end-effector pose, gripper aperture, and force sensing. Based on these modalities, it segments the robot trajectory into a sequence of manipulation phases. At each phase, Robo2VLM uses scene and interaction understanding to identify 3D properties of the robot, task goal, and the target object. The properties are used to generate representative VQA queries – images with textural multiple-choice questions – based on spatial, goal-conditioned, and interaction reasoning question templates. We use a subset of Open X-Embodiment to generate Robo2VLM-1, a large-scale in-the-wild dataset with 684,710 questions based on 463 distinct scenes and 3,396 robotic manipulation tasks from 176k real robot trajectories. Results suggest that Robo2VLM-1 can benchmark and improve VLM capabilities in spatial and interaction reasoning.

## 1   Introduction

Emerging Vision-Language Models (VLMs) [1, 2, 3, 4, 5, 6, 7] can perform high-level reasoning and scene interpretation [8, 9]. Recent robotic manipulation systems that integrate VLMs demonstrate enhanced capabilities in semantic and long horizon task reasoning [10, 11, 12]. Yet, *the* key challenge persists: the image-text corpora used for VLM pre-training high-quality lack fine-grained spatial information, which are prerequisites for robots to identify long-tail objects, complex scenes, reason about spatial relationships, and plan physical interactions.

To address this challenge, some research [13, 14, 15] relies on data generation through simulation [16, 17, 18]. However, such data has inherent limitations due to the sim-to-real gap, because simulator cannot accurately model visual properties such as noise, clutter, and lighting variations and physical properties such as contact dynamics, and interactions. Therefore, strong performance in simulation often fails to translate reliably to the physical world. Meanwhile, deriving spatial knowledge from real-world ("in-the-wild") data typically requires extensive and costly human labeling [19, 20]. In contrast, teleoperated robot trajectories that are used to train visuomotor policies [21], such as Vision-Language-Action(VLA) [10, 22] or diffusion policies [23], typically include precise, structured

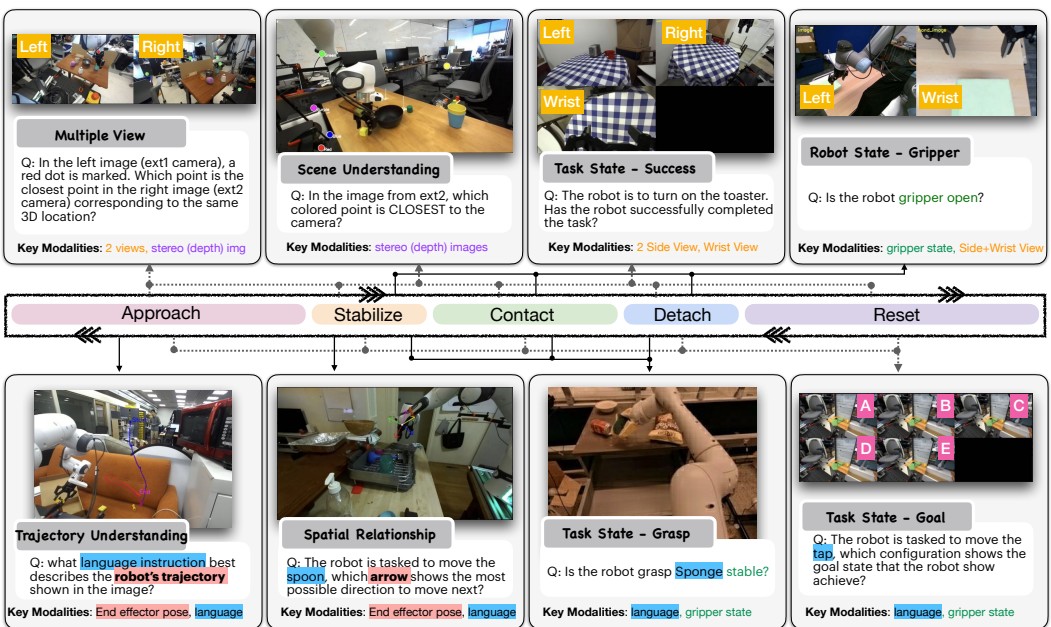

Figure 1: **Robo2VLM-1 dataset overview**. The middle colorbar traces a typical manipulation episode—from pre-grasp through immobilization, contact, detach, and into post-grasp. Surrounding panels give example questions for each VQA category. Dashed arrows connect every category to the phase(s) in which its questions are sampled. Icons beneath each panel list the key sensing modalities (RGB, stereo depth, wrist/side cameras, gripper state, end-effector pose, language instructions) needed to derive ground-truth answers.

proprioceptive and kinematic information, such as joint angles, end-effector poses, gripper states, and force–torque readings, that implicitly encode 3D spatial information. We hypothesize that visual and textual data extracted from robot trajectories can improve VLM's spatial reasoning capabilities.

We present Robo2VLM, a multiple-choice Visual Question Answering (VQA) dataset generation framework for VLMs from real-world robot data. Given a human-teleoperated robot trajectory, Robo2VLM segments the trajectory into distinct manipulation phases, selects representative frames from each phase, and generates questions whose answers are supported by synchronized proprio-ceptive and kinematic ground truth. We apply Robo2VLM to 176k diverse, real-world trajectories from the Open X-Embodiment (OXE) dataset [24], producing over 3 million VQA samples. Inspired by data optimization paradigms such as domain reweighting in natural language processing [25] and robot policy learning [26], we curate Robo2VLM-1, a large-scale, in-the-wild VQA dataset with 684,710 questions covering 463 distinct scenes, 3,396 robotic manipulation tasks, and 149 manipulation skills.

We evaluate 14 model configurations with state-of-the-art open source models (LLaVA, Llama and Qwen) and with different parameter sizes and prompting techniques. The results indicate that some VLMs can achieve near human performance in questions related to object reachability and interaction understanding. Evaluation also suggests a significant gap to human performance, especially in complex reasoning of fine-grained spatial relationship and interactions. Finetuning LLaVA [4] with Robo2VLM-1 improves most of the spatial and interaction capabilities with increasing training dataset size, with a maximum 50% accuracy gain in state reasoning and task understanding.

This paper makes the following contributions: (1) Robo2VLM, a VQA data generation framework from real robot trajectories. (2) Robo2VLM-1, an open VQA dataset with 684,710 questions covering diverse and realistic evaluation scenarios for manipulation. (3) Extensive evaluation data on state-of-the-art and fine-tuned VLMs.

## 2 Related Work

**Large-Scale Robotics Datasets** Recent large-scale robotics datasets, such as Open-X-Embodiment [24] and DROID [27], provide extensive teleoperated demonstrations

of complex manipulation skills. These datasets are foundational for training modern generalist robot policies—including Octo [22], RT-1 [28], RT-2 [29], OpenVLA [10], Gemini Robotics [11], $\pi_0$ [30], and Hi Robot [12]—enabling them to learn diverse skills and understand nuanced physical interactions from broad data. Crucially for grounding VLMs, robotics datasets from Open-X-Embodiment contains rich sensory-modal including RGB video, proprioceptive [31, 28, 32, 33, 34, 35, 36, 37, 38, 39, 40, 41, 42, 43, 44], depth data [31, 28, 32, 33, 35], and force-torque [37, 39, 40, 41], that reflect the dynamics of interaction. These information presents an opportunity to bridge robotics data with VLMs.

**VQA Benchmarks for Robotics and Embodied AI**  VQA offers a powerful paradigm for evaluating the visual reasoning capabilities of VLMs [45, 46, 47]. Recently, VQA benchmarks have been developed for robotic tasks such as visual navigation in long-horizon planning [48, 49]. Simulation-based approaches [13, 14, 15] (often utilizing environments like [16, 17, 18]) generate large-scale VQA dataset, but face the persistent sim-to-real domain gap, where the result may not hold in reality due to factors like noise, clutter, and lighting variations. Real-world data benchmark, such as RoboVQA [**?** ] (human-verified Q/A), improve generalization to real world setting but often involve significant manual annotation effort. These methods typically do not fully automate VQA generation by exploiting the rich spectrum of non-visual modalities (e.g., force, torque, proprioception), limiting their ability to support questions grounded in concepts such as grasp stability or multi-view spatial alignment. In contrast, Robo2VLM reduces the need for manual annotation and enables interaction and physical properties reasoning that are underexplored in previous VQA benchmarks, such as gripper states, grasping stability, task goal, and spatial information focus on the robot and target objects.

## 3 Robo2VLM

Robo2VLM generates five-way multiple-choice question answering (MCQ) from real robot teleoperated trajectories. Robo2VLM offers the following key features: (1) High-quality and representative keyframe selection from long-horizon, in-the-wild, multi-modal robot trajectories, ensuring semantic diversity and relevance; (2) Manipulation-centric question generation encompassing spatial, goal-conditioned, and interaction reasoning, each aligned with specific manipulation phases and grounded in corresponding sensor modalities.

We begin by defining a robot trajectory as a time-synchronized sequence of data frames from multiple sensor modalities including exteroceptive and proprioceptive [50]. Let $T$ denote the length of a trajectory, and let $t \in \{1, 2, \ldots, T\}$ index the discrete time steps.

**Definition 3.1** *(Robot Observation Data Frame)* At each time step $t$, the robot data frame is represented as a tuple:

$$\mathcal{D}_t = \left( \mathcal{I}_t^{\text{RGB}}, \mathcal{I}_t^{\text{Stereo}}, \mathbf{p}_t^{\text{EE}}, s_t^{\text{Gripper}}, \mathbf{f}_t \right)$$

where $\mathcal{I}_t^{\text{RGB}} = \{I_t^{\text{RGB}} \in \mathbb{R}^{H \times W \times 3}\}$ is a set of multi-view RGB images captured from monocular cameras, $\mathcal{I}_t^{\text{Stereo}} = \{I_t^{\text{Stereo}} \in \mathbb{R}^{2 \times H \times W \times 3}\}$ denotes a set of multi-view stereo image pair (left and right) if available, $\mathbf{p}_t^{\text{EE}} \in SE(3)$ is the 6-DoF end-effector pose and $s_t^{\text{Gripper}} \in \mathbb{R}$ denotes the scalar gripper state such as gripper aperture, $\mathbf{f}_t \in \mathbb{R}^6$ is the force-torque vector from the end-effector sensor.

The camera images are referred as exteroceptive sensing and the end-effector-related states belong to proprioceptive sensing.

**Definition 3.2** *(Robot Trajectory)* A trajectory $\mathcal{T}$ is defined as the temporally ordered sequence of observations $\mathcal{D}_{1:T}$ with a trajectory task language description $l$:

$$\mathcal{T} = \{\mathcal{D}_{1:T}, l\}$$

Given a robot trajectory, Robo2VLM (Fig. 2) begin with *scene-interaction understanding*, applying semantic segmentation and manipulation phase classification to identify key segments (e.g., pre-grasp/approaching, contact, grasp, release). From these, we extract *keyframes* based on phase transitions, scene coverage, and visibility of objects or the robot across multiple camera views. We use manipulation domain knowledge to design *question prototype* to target core manipulation skills

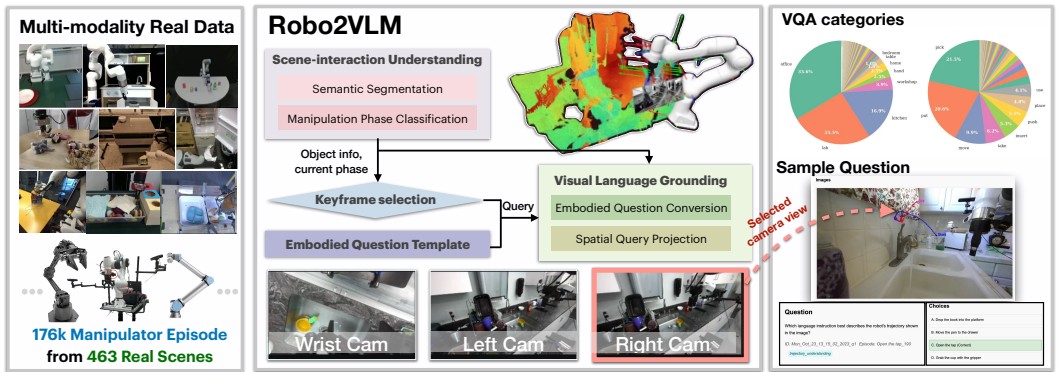

Figure 2: **Robo2VLM framework.** Robo2VLM generates multi-modal real-world robot trajectories through (1) manipulation phase classification, (2) keyframe selection guided by scene and interaction cues, and (3) structured VQA question prototype.

such as spatial relationship, goal conditions, and interaction understanding. Robo2VLM instantiates these prototypes on selected keyframes and transforms them into natural language multiple-choice questions via a *visual-language grounding* module that performs question conversion and spatial query projection.

### 3.1 Scene-Interaction Understanding

**Embodied Scene Understanding**   Given a task description in nature language and all images from different camera views, we first parse the language instruction using an off-the-shelf LLM such as Qwen 2.5 [2] to obtain {target object}, scene, task, and skill description. For the spatial understanding in manipulation, we need to know the relative direction and displacement between target object and gripper. From the proprioceptive data, we obtain the target object interaction point ground-truth from the robot trajectory data frames.

**Manipulation Phase Segmentation**   To segment robotic manipulation trajectories into semantically meaningful phases [51], we define a temporal phase classification function based on the sequence of end-effector poses, gripper aperture signals, and force-torque measurements: $\mathbf{p}_{1:T}^{\mathrm{EE}}, s_{1:T}^{\mathrm{Gripper}}, \mathbf{f}_{1:T}$. To align different types of gripper aperture, $s_t^{\mathrm{Gripper}}$ is normalized to $[0, 1]$, where 0 indicates fully open and 1 indicates fully closed. Let $s_t \in [0, 1]$ denote the normalized aperture at time $t$, and $\Delta s_t = s_t - s_{t-1}$ its temporal derivative. $\Delta s_t \approx 0$ denotes a small change within a tolerance margin $\epsilon$, typically set to filter out noise. Let $\|\mathbf{f}_t\|$ be the force magnitude (if available). We introduce three threshold parameters: $\tau_g$ (grasp threshold), $\tau_c$ (closure threshold), and $\tau_f$ (force threshold for contact detection). Manipulation processes can be represented as a sequence of discrete phases, including approaching, stabilizing, contacting, releasing, and resetting or transitioning to subsequent actions. We denote the phase varible as $\Phi = \{\Phi_{\mathrm{app}}, \Phi_{\mathrm{stab}}, \Phi_{\mathrm{cont}}, \Phi_{\mathrm{rel}}, \Phi_{\mathrm{reset}}, \Phi_{\mathrm{trans}}\}$. Each timestep $t$ is assigned a label $\phi_t \in \Phi$ according to the following temporal logic rules:

$$\phi_t = \begin{cases} \Phi_{\mathrm{app}} & \text{if } s_t < \tau_g \wedge \Delta s_t < -\epsilon \\ \Phi_{\mathrm{stab}} & \text{if } \phi_{t-1} = \Phi_{\mathrm{app}} \wedge s_t < \tau_g \wedge |\Delta s_t| \leq \epsilon \\ \Phi_{\mathrm{cont}} & \text{if } \phi_{t-1} = \Phi_{\mathrm{stab}} \wedge s_t \geq \tau_c \wedge |\Delta s_t| \leq \epsilon \wedge (\|\mathbf{f}_t\| > \tau_f \vee \text{force unavailable}) \\ \Phi_{\mathrm{rel}} & \text{if } \phi_{t-1} = \Phi_{\mathrm{cont}} \wedge s_t \geq \tau_c \wedge \Delta s_t > \epsilon \\ \Phi_{\mathrm{reset}} & \text{if } \phi_{t-1} = \Phi_{\mathrm{rel}} \wedge s_t < \tau_g \wedge \Delta s_t > \epsilon \\ \Phi_{\mathrm{trans}} & \text{otherwise} \end{cases}$$

The inclusion of force magnitude ensures that passive closure without external contact is not misclassified as active interaction. This multimodal phase labeling strategy captures both kinematic intent and physical contact, enabling robust segmentation of diverse manipulation behaviors.

To enforce a temporally coherent yet flexible phase progression, we define a partial order over the manipulation phases:

$$\Phi_{\mathrm{app}} \prec \Phi_{\mathrm{stab}} \prec \Phi_{\mathrm{cont}} \prec \Phi_{\mathrm{rel}} \prec \Phi_{\mathrm{reset}} \to \Phi_{\mathrm{app}}$$

Table 1: Categorization of visual reasoning questions for robotic manipulation, with manipulation phase (color-coded) and data modality context. ▢ Approach, ▢ Stabilize, ▢ Contact, ▢ Release, ▢ Rest.

| Capabilities | Question Prototype | Manip. Phase | Sensor Modality |
|---|---|---|---|
| **Spatial Reasoning** | | | |
| Object State | Is the {target object} reachable by the robot? | Approach | $I_t^{\text{RGB}}$, $D_t$ |
| Spatial Relationship | What's the relative direction in 3-D between end effector and {target object}? | Approach, Stabilize | $I_t^{\text{RGB}}$, $\mathbf{p}_t^{\text{EE}}$ |
| Scene Understanding | Which point is closer to the camera viewing the scene? | Approach, Stabilize | $I_t^{\text{RGB}}$, $I_t^{\text{Stereo}}$ |
| Multiple View | Which point in the right-side image corresponds to the point in the left-side image? | Approach, Stabilize, Release, Rest | $I_t^{\text{Stereo}}$ |
| **Goal-conditioned Reasoning** | | | |
| Task State-success | Has the robot successfully completed the task? | Rest | $I_t^{\text{RGB}}$ |
| Task State-Goal | What is the goal configuration for {interaction}? | Stabilize, Contact, Release | $I_t^{\text{RGB}}$, $\mathbf{p}_t^{\text{EE}}$ |
| Action Understanding | The robot is {interaction}. What is the robot's current action phase? | Approach, Stabilize, Contact, Release, Rest | $I_t^{\text{RGB}}$, $\mathcal{T}_{1:t}$ |
| Interaction Phase | What will the robot do next? | Approach, Stabilize, Contact, Release | $I_t^{\text{RGB}}$, $\dot{\mathbf{p}}_t^{\text{EE}}$ |
| Trajectory Understanding | What task does this trajectory likely accomplish? | Approach | $I_t^{\text{RGB}}$, $\mathbf{p}_t^{\text{EE}}$ |
| **Interaction Reasoning** | | | |
| Task State-grasp | Is this a stable grasp? | Stabilize, Contact, Release | $I_t^{\text{RGB}}$, $\mathbf{f}_t$ |
| Robot State | Is the robot gripper currently open? | Stabilize, Contact, Release | $I_t^{\text{RGB}}$, $s_t^{\text{Gripper}}$ |

This structure enforces unidirectional transitions along the phase chain, while allowing both phase skipping (e.g., directly from $\Phi_{\text{app}}$ to $\Phi_{\text{cont}}$) and looping from the terminal phase $\Phi_{\text{reset}}$ back to the initial phase $\Phi_{\text{app}}$, which is common in sequential manipulation routines. At each time step $t$, the phase label must satisfy $\phi_t \succeq \phi_{t-1}$, or $\phi_t = \Phi_{\text{app}}$ if $\phi_{t-1} = \Phi_{\text{reset}}$, ensuring temporal monotonicity or task repetition without reversal. The auxiliary state $\Phi_{\text{trans}}$ is used for ambiguous, missing, or conflicting observations where no confident assignment is possible. This symbolic, temporally-constrained model supports robust segmentation of complex manipulation behaviors under noisy or partially missing sensory input.

## 3.2 Visual Question Prototypes

We design a set of *visual question prototypes*, each of which aligns with specific manipulation task completion required robot capabilities and anchors to distinct manipulation phases as illustrated in Table 1. These prototypes are organized into three reasoning categories.

**Spatial Reasoning** focuses on the robot's understanding of object geometry, reachability, and spatial layout across viewpoints. Questions such as "Is the object reachable?" or "What's the relative direction between the gripper and the object?" are grounded in the early approach ▢ and stabilize ▢ stages. These rely on RGB, depth, stereo, and 3D gripper pose data, which together enable accurate localization and spatial inference across frames or views.

**Goal-conditioned Reasoning** probes the agent's high-level understanding of tasks, including goal inference, future action prediction, and overall task success. Questions such as "Is the task failed?", "What will the robot do next?", and "What is the robot's current action phase?" span multiple manipulation phases from approach ▢ through reset ▢. These require temporal context, pose estimation, and sometimes motion history, leveraging the multi-step evolution of the scene.

**Interaction Reasoning** focuses on physical interaction dynamics, such as grasp stability or the robot's current actuator state. These occur during stabilize ▢, contact ▢, and release ▢ phases, and depend on RGB, tactile, or gripper aperture signals. For instance, the question "Is this a stable grasp?" may depend on contact force readings or inferred object displacement.

The ground truth of the questions are grounded by multiple sensor modality observations. We design the incorrect answers as part of the visual question prototypes. For example, in the scene understanding, we require the sampled points to be significantly different in depth from other points and from the depth sensor to account for sensor inaccuracy. In action understanding, the correct action arrow differs significantly from the distractor arrows by having a large angular separation in the projected 2D image. To detect guessing by hallucination, we randomly replace some correct answers with "None of Above" option.

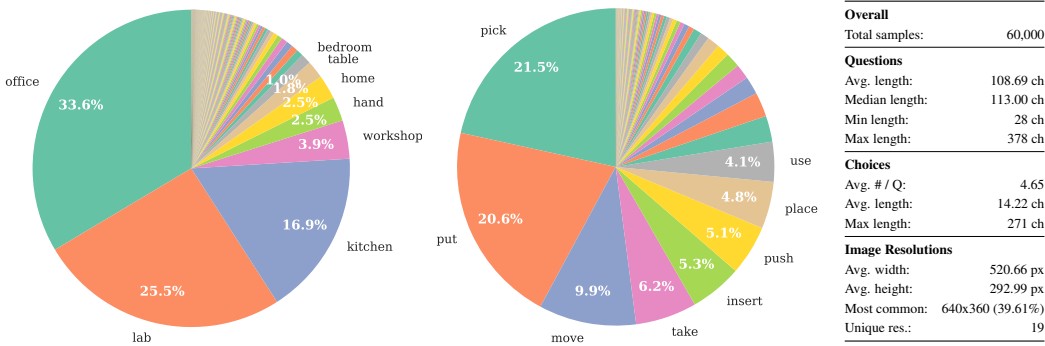

Figure 3: **Distribution and key statistics of Robo2VLM-1 dataset.** (Left) Robo2VLM-1 covers diverse scenes with the most frequent scenes in office (33.6%), lab (25.3%), and kitchen (16.9%). (Middle) Robo2VLM-1 covers tasks including common manipulation actions include pick (21.5%), put (20.6%), and move (9.9%). (Right) The table summarizes key dataset statistics including question characteristics, answer choices, and image resolutions.

## 3.3 Keyframe Selection

Given that raw robotic trajectories often contain hundreds of frames sampled at high frequency, using all frames is computationally expensive and can introduce redundancy due to minimal temporal variation. Moreover, many intermediate frames are visually or semantically uninformative for downstream reasoning tasks. To address this, we select a compact set of keyframes that retain essential semantic and visual cues while reducing redundancy and data volume. These keyframes are extracted from the multi-modal robot trajectory $\mathcal{T} = \{\mathcal{O}_t\}_{t=1}^{T}$ based on manipulation phase transition, scene coverage diversity and context visibility.

## 4 Robo2VLM-1 Dataset

**Open X-Embodiment and its datasets** Open X-Embodiment (OXE) [24] is major collaborative research initiative that aggregates robotic demonstration data collected from 22 different robot embodiments across 35 research labs worldwide, encompassing over 1 million trajectories covering more than 500 skills. Applying domain reweighting [25], we select a subset focusing on manipulation with real robot embodiments. In total, we use 13 datasets [31, 29, 32, 33, 34, 35, 36, 37, 39, 40, 41, 42, 43, 38, 44] with a total of 176,139 trajectories. While most modalities are included in Open X-Embodiments release, we manually include modalities introduced by the original paper. For example, DROID dataset [31] includes camera calibration information and stereo depth. The detailed modality inclusion can be found in Table. 2.

Table 2: Trajectories and sensing modalities across datasets with a total of 176k trajectories. **# Traj**: number of trajectories; **Prop**: joint-state proprioception; **Dpth**: depth images; **GripAp**: gripper-aperture signal; **# VQA**: number of questions. ✓ denotes modality is available, ✗ denotes absent.

| Dataset | # Traj | Prop | Dpth | GripAp | # VQA |
|---|---|---|---|---|---|
| DROID [31] | 92k | ✓ | ✓ | ✓ | 299k |
| Fractal [28] | 73k | ✓ | ✗ | ✓ | 267k |
| Kuka MM [34] | 3k | ✓ | ✓ | ✓ | 25k |
| Autolab [35] | 896 | ✓ | ✓ | ✓ | 22k |
| Sirius [36] | 600 | ✓ | ✗ | ✓ | 21k |
| MVP [37] | 480 | ✓ | ✗ | ✓ | 8k |
| VINN [38] | 435 | ✗ | ✗ | ✗ | 34 |
| Fanuc [39] | 415 | ✓ | ✗ | ✓ | 11k |
| TableTop [41] | 110 | ✓ | ✗ | ✓ | 5k |
| VIOLA [42] | 135 | ✓ | ✗ | ✓ | 8k |
| BUDS [43] | 50 | ✓ | ✗ | ✓ | 6k |
| ROT [44] | 14 | ✓ | ✗ | ✓ | 245 |

**Robo2VLM for Open X-Embodiment** We use Robo2VLM to process selected robot demonstrations from the Open X-Embodiment dataset by selecting and interpreting the scenes. The entire process takes 2935.7 GPU hours on Nvidia A100 GPUs. For each selected keyframe, Robo2VLM instantiates questions from embodied question templates resulting in the generation of a pool of over 3 million VQA question-answer pairs on all robot demonstrations. The initial pool is imbalanced due to the availability of different data sources.

Table 3: Performance Comparison of Multimodal Foundation Models on OpenX-VQA Benchmark Categories (%). Upper part: zero-shot. Lower part: with CoT prompting.

| Model | Overall | Spatial Reasoning | | | | | Goal Reasoning | | | Interaction Reasoning | | |
|---|---|---|---|---|---|---|---|---|---|---|---|---|
| | | RS | OS | SR | SU | MV | TS-G | TS-S | TS-GL | AU | IP | TU |
| | (%) | (%) | (%) | (%) | (%) | (%) | (%) | (%) | (%) | (%) | (%) | (%) |
| *Zero-Shot* | | | | | | | | | | | | |
| LLaVA 1.5-7B | 21.58 | 35.32 | 23.87 | 16.08 | 17.78 | 17.50 | 31.82 | 23.79 | 19.03 | 20.30 | 21.74 | 22.37 |
| LLaVA 1.6 Mistral-7B | 24.09 | 30.31 | 35.13 | 19.42 | 20.24 | **19.29** | 34.20 | 30.77 | **19.52** | 18.67 | 20.70 | 22.83 |
| LLaVA 1.6-34B | 24.94 | 26.66 | 29.75 | 21.47 | 23.18 | 17.86 | 29.19 | 29.40 | 17.90 | 19.49 | 36.98 | 30.59 |
| Llama 3.2-90B | 28.60 | 31.94 | 55.87 | 18.51 | 26.61 | 16.43 | 28.23 | 35.27 | 8.06 | 18.13 | 51.56 | 49.77 |
| Qwen 2.5 VL-7B | 30.63 | 41.68 | 55.63 | 21.55 | 24.38 | 17.32 | 33.01 | 42.57 | 7.82 | 25.71 | 46.61 | 39.73 |
| Qwen 2.5 VL-32B | 37.68 | **49.39** | 71.37 | 21.85 | **28.53** | 17.50 | **34.21** | 55.08 | 12.90 | 30.45 | 63.80 | 49.32 |
| Qwen 2.5 VL-72B | **37.76** | 38.84 | **85.00** | 22.31 | 28.23 | 15.71 | 28.47 | 51.89 | 10.08 | **33.96** | 71.09 | **54.79** |
| *CoT Reasoning* | | | | | | | | | | | | |
| LLaVA 1.5-7B | 21.61 | 28.28 | 21.00 | 17.37 | 20.90 | 18.93 | 25.36 | 24.19 | **21.53** | 21.24 | 20.31 | 20.09 |
| LLaVA 1.6 Mistral-7B | 24.05 | 27.60 | 38.87 | 17.15 | 20.18 | **22.32** | 25.84 | 28.03 | 18.47 | 18.40 | 30.60 | 29.68 |
| LLaVA 1.6-34B | 23.49 | 20.43 | 31.00 | 21.24 | 22.88 | 20.36 | 18.18 | 26.14 | 16.77 | 21.79 | 35.16 | 26.94 |
| Llama 3.2-90B | 30.45 | 32.34 | 79.87 | 13.35 | 26.37 | 18.57 | 29.90 | 29.14 | 14.27 | 19.76 | 59.24 | 44.75 |
| Qwen 2.5 VL-7B | 34.82 | 38.02 | 90.00 | **21.78** | 23.30 | 16.79 | **36.84** | 46.48 | 18.39 | 28.15 | 42.71 | 36.99 |
| Qwen 2.5 VL-32B | **41.30** | 48.85 | 90.50 | 18.82 | 29.19 | 19.82 | 35.17 | **60.43** | 18.71 | 32.21 | 71.35 | **49.32** |
| Qwen 2.5 VL-72B | 39.52 | 44.79 | **92.37** | 18.36 | **29.73** | 13.39 | 29.19 | 55.28 | 13.15 | **36.13** | 74.09 | 46.12 |

*Category Abbreviations:* **Spatial Reasoning:** RS: Robot State (gripper/arm position estimation), OS: Object State (object reachability/manipulability), SR: Spatial Relationship (relative positioning between robot and objects), SU: Scene Understanding (spatial layout comprehension), MV: Multiple View (cross-view correspondence). **Goal-Conditioned Reasoning:** TS-G: Task State-grasp (grasp stability assessment), TS-S: Task State-success (task completion status), TS-GL: Task State-goal (goal configuration understanding), **Interaction Reasoning:** AU: Action Understanding (robot's current action phase), IP: Interaction Phase (prediction of next robot action), TU: Trajectory Understanding (overall task interpretation).

**Robo2VLM-1 Curation**  Inspired by data optimization paradigms such as domain reweighting in natural language processing [25] and robot policy learning [26], our curation process aims to balance the distribution of questions across diverse scene and task types. It selects a representative and high-quality subset of questions that effectively balances diversity across scenes, tasks, skills, and reasoning types, while ensuring clarity and unambiguous ground truth. In total, Robo2VLM-1 contains 684,710 questions, spanning 463 distinct real-world scenes, 3,396 unique robotic manipulation tasks, and 149 different manipulation skills.

## 5 Experiment

In this section, we sample 60k VQA from Robo2VLM-1 with a 50k training set (Robo2VLM-1-Train) and a 10k testing set (Robo2VLM-1-Test). We mainly study two research questions: (1) How does Robo2VLM-1-Train dataset improve the spatial and interaction reasoning capabilities of VLMs? and (2) How effectively does Robo2VLM-1-Test evaluate VLMs in these reasoning tasks?

**Evaluation Setup**  We benchmark state-of-the-art open-source models in different configurations, including LLaVA, Llama 3.2 Vision, and Qwen2-VL/Qwen2.5-VL. Each model is evaluated under both zero-shot and Chain-of-Thought (CoT) prompting settings. For CoT, we follow the prompting strategy from [11] by appending the following instruction to the end of each question: *"Reason step by step about the answer, and show your work, for each step. Only after that, proceed to the final answer."* We run a simultaneous Llama-3.2-3B-Instruct to extract model outputs for final letter answer. We focus fine-tuning on language layers (both attention and MLP modules) while keeping vision layers frozen. For each configuration, we use random 2000 questions from the testing set. For consistency, all models are evaluated with a temperature of 0.7, a maximum completion token length of 4096, and overall context length of 10240. All models use their vision or vision instruct version with float16 quantization. All models are evaluated with 8 Nvidia A100 GPUs with 80GB memory. We use LoRA to fine-tune LLaVA 1.6 with rank 128 and alpha 256.

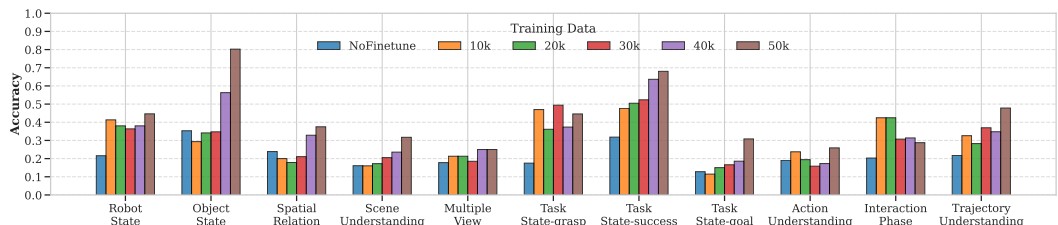

Figure 4: **Fine-tuning LLaVA 1.6 with increasing training data of Robo2VLM-1** from 10k to 50k VQA items. Accuracy improvements almost all categories compared to no fine-tuning.

## 5.1 Benchmark with Robo2VLM-1

Table 3 presents a detailed comparison of vision–language foundation models on the Robo2VLM-1 benchmark, evaluated under both zero-shot and Chain-of-Thought (CoT) prompting conditions. The results reveal nuanced interactions across model architecture, scale, and reasoning strategy.

**Cross-Model Performance:** Evaluation data on Robo2VLM-1-test suggests that Qwen models has higher overall accuracy compared to other VLMs of the same configuration, which align with the observation from other VQA benchmarks such as [52, 53]. Qwen 2.5 VL-72B achieves the highest zero-shot accuracy at 37.76%, while Qwen 2.5 VL-32B achieves 41.30% overall accuracy in the CoT setting. Qwen models particularly excel in object-centric categories such as Object State, where Qwen 2.5 VL-72B reaches 85.00% (zero-shot) and 92.37% (CoT), and Interaction Phase (IP) (71.09% zero-shot, 74.09% CoT for 72B).

**Impact of Model Scale.** Zero-shot accuracy generally improves with model size — rising from 30.63% (Qwen 7B) to 37.76% (Qwen 72B). However, this trend does not hold in the CoT setting, where the 32B model surpasses the 72B model (41.30% vs. 39.52%). The observation aligns the official technical report of Qwen2.5[2] that the mathematical and problem-solving capabilities of Qwen2.5-VL-32B are further enhanced through reinforcement learning. LLaMA models display a different trend — while the 11B model outperforms the 90B version in zero-shot setting, the larger model benefits more under CoT prompting, suggesting that scaling may unlock latent capabilities only when paired with explicit reasoning support.

**Effectiveness of CoT Prompting:** CoT prompting generally enhances performance for both Qwen and LLaMA models. For example, Qwen 2.5 VL-7B improves from 30.63% to 34.82%, and LLaMA 3.2-90B increases from 28.60% to 30.45%. The most substantial gains are observed in Qwen 2.5 VL-32B, which improves from 37.68% to 41.30%. Results suggest that CoT benefits Task State–Success(from 55.08% to 60.43%), and Interaction Phase (from 63.80% to 71.35%). However, in the Spatial Relationship category, for example, Qwen 32B's accuracy drops from 21.85% to 18.82%, indicating that verbose reasoning chains may introduce noise in tasks requiring precise spatial localization.

## 5.2 Finetuning with Robo2VLM-1

We perform model finetuning experiment using Robo2VLM-1-train and evaluate on Robo2VLM-1-test. We increase the training data samples from 10k to 50k in finetuning. As depicted in Figure 4, increasing the fine-tuning data generally leads to notable performance enhancements across most VQA categories. Significant gains are observed in 'Object State' understanding, where accuracy improved from 29.34% to 80.24%. "Task State-success" also sees a substantial rise from 47.65% to 68.03%. Other categories demonstrating clear positive trends with more data. However, in some categories such as Spatial Relationship and Task State–Goal, fine-tuning with limited data (e.g., 10k) underperforms the no-finetuning baseline. This may be because the model has not yet seen enough task-specific examples to begin generalizing, or because the question formats in Robo2VLM-1 differ from those seen during pretraining, requiring adaptation time. In some categories, finetuning with Robo2VLM-1 does not improve the performance due to the reasoning capability limitation of the base model. This is also reflected in the fact that LLaVA shows performance degradation in CoT prompting in Table 3. The "interaction phase" question requires the model to predict the next frame,

demanding complex reasoning and making it a particularly challenging problem. This suggests that for complex tasks, the base model language performance is important for further improvement with Robo2VLM-1.

## 5.3 Comparison with Human Performance

We conducted a human evaluation covering all 11 categories defined in Table 3. For each category, a human evaluator was asked to randomly answer questions from Robo2VLM-1-test. We use the average success rate as a reference for comparison with three models—LLaVA 1.6-7B, LLaVA 1.6-7B-Finetuned, and Qwen 2.5 VL-32B—CoT on the same set of categories as shown in Figure 5. Qwen 2.5 VL-32B—CoT achieves near human accuracy, with 90.5% in Object State compared to 96.7% for humans, and 71.35% in Interaction Phase versus the human score of 80.0%. In more complex spatial reasoning tasks such as Spatial Relationship, where human achieves 60.0% accuracy, the best model (LLaVa 1.6-7B, finetuned) reaches only 19.42%. This may suggest that even if observing from multiple views, a monocular image may lack the full depth information needed to accurately determine the spatial relationship. Furthermore, finetuning enhances model performance. LLaVA 1.6-7B finetuned on the Robo2VLM-1 training dataset shows consistent improvements across multiple categories, particularly in Task State, Object State, and Trajectory Understanding,

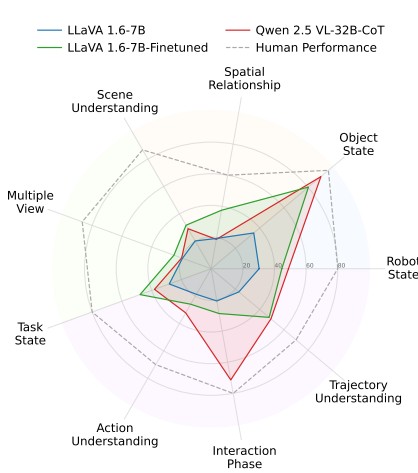

Figure 5: Comparison of human performance to different multimodal foundation models.

compared to its non-finetuned LLaVA 1.6-7B. These findings demonstrate the potential Robo2VLM-1 in studying and narrowing the gap between model and human performance in spatial and task reasoning.

## 6 Conclusion and Discussion

In this paper, we introduce Robo2VLM, a framework that generates VQA grounded in robot sensory modalities. We apply Robo2VLM to 176k real robot trajectories from Open X-Embodiment, and curate Robo2VLM-1, a comprehensive dataset of 684,710 questions covering 463 distinct scenes, 3,396 robotic manipulation tasks, and 149 manipulation skills. Evaluation of state-of-the-art open-source VLMs suggests that some VLMs, such as Qwen2.5 VL 32B with CoT prompting, can achieve near human performance in questions related to object reachability and interaction understanding, while there is a significant gap to human in reasoning fine-grained spatial relationship and interactions. Evaluation also suggests that finetuning Robo2VLM-1 dataset improves in spatial and interaction reasoning. Future work will focus on generalizing Robo2VLM to a wider range of robot embodiments and generating an even more diverse dataset. We also plan to explore the deployment of models trained on Robo2VLM-1 to real-world robotic tasks.

**Limitations** We acknowledge that Robo2VLM is a data generation framework that relies on the quality of input tele-operated trajectories. If the original trajectory is wrongly calibrated, it compromises the quality of generated VQA data. Or if the original trajectory misses embodiment sensory modalities, such as NYU VINN [38] (0.2% of the 176k trajectories), it limits the amount of questions that Robo2VLM can generate. Additionally, because Robo2VLM constructs question–answer pairs from templates, it needs additional mechanisms to introduce linguistic diversity.

## Acknowledgement

This research was performed at the AUTOLAB at UC Berkeley in affiliation with the Berkeley AI Research (BAIR) Lab. This work is supported in part by donations from Google.

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

## A   Broader Impact

The development of Robo2VLM and Robo2VLM-1 aims to accelerate progress in robotic manipulation by providing a robust framework for evaluating and improving Vision-Language Models. Positive societal impacts are significant. More capable robots, enhanced by VLMs rigorously tested on such benchmarks, can revolutionize various sectors. In manufacturing, they can lead to more efficient, flexible, and safer production lines by undertaking complex assembly or hazardous material handling. In healthcare, advanced robotic assistants could support surgeons with greater precision, provide personalized care for the elderly or individuals with disabilities, thereby improving their quality of life and independence, and assist in laboratory automation for faster medical research. For domestic tasks, robots could alleviate household burdens, freeing up human time for more creative or relational pursuits. Beyond these, such advancements can contribute to safer work environments by automating dangerous jobs in construction, mining, or disaster response, and even aid in environmental conservation efforts through automated monitoring and intervention. The increased productivity and innovation spurred by these technologies could lead to economic growth and the creation of new job categories focused on designing, maintaining, and overseeing these intelligent systems. However, it is important to consider potential negative societal impacts. As VLMs become more powerful through evaluation on such benchmarks, there's a risk of misuse if these capabilities are applied to autonomous systems without appropriate safeguards, potentially leading to unintended actions or job displacement in certain sectors. For example, if the underlying trajectory data in Robo2VLM inadvertently contains biases (e.g., related to specific environments, objects, or human demonstrators), models trained or evaluated on Robo2VLM-1 might perpetuate or amplify these biases. Future work should actively consider methods to detect and mitigate such biases in the dataset and the models. Furthermore, while the goal is to advance AI for beneficial applications, any significant improvement in generative or understanding capabilities of models could, in principle, be adapted for unintended purposes. Therefore, ongoing discussion and development of ethical guidelines and safety protocols are crucial as VLM capabilities advance in robotics and other fields.

## B   Question Analysis

The complete dataset can be found in the huggingface website, https://huggingface.co/datasets/keplerccc/Robo2VLM-1. We provide representative examples to show the diversity and quality of the dataset. Each VQA contains one/multiple images showing the robot current position and the scene, a language description question, and multiple choices as candidate answer.

### B.1   Example Questions from Different Tasks

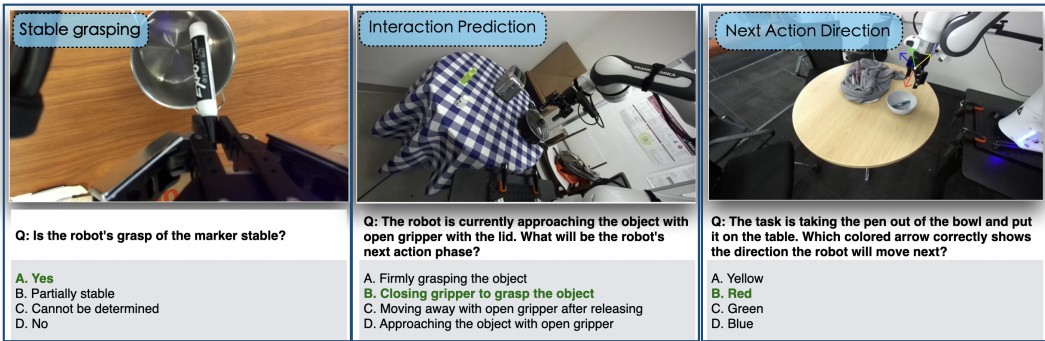

Figure 6: **Example VQAs**. Each panel illustrates a distinct category of visual question answering grounded in real robot interactions.

The examples in Figs. 6,7 highlight the diversity and complexity of visual question answering (VQA) tasks grounded in real-world robotic manipulation. Each question may be associated with multiple images, which can originate from different phases of the manipulation sequence or from distinct camera viewpoints. This design reflects the inherently temporal and multi-perspective nature of robotic tasks, requiring models to reason over a sequence of actions or fuse complementary

observations. The questions span reasoning types such as goal configuration prediction, task outcome evaluation, grasp stability assessment, and interaction phase forecasting. These diverse formats challenge models to integrate spatial understanding, temporal progression, and multimodal cues, making the dataset a rigorous benchmark for evaluating the task-level reasoning capabilities of vision-language models in robotics.

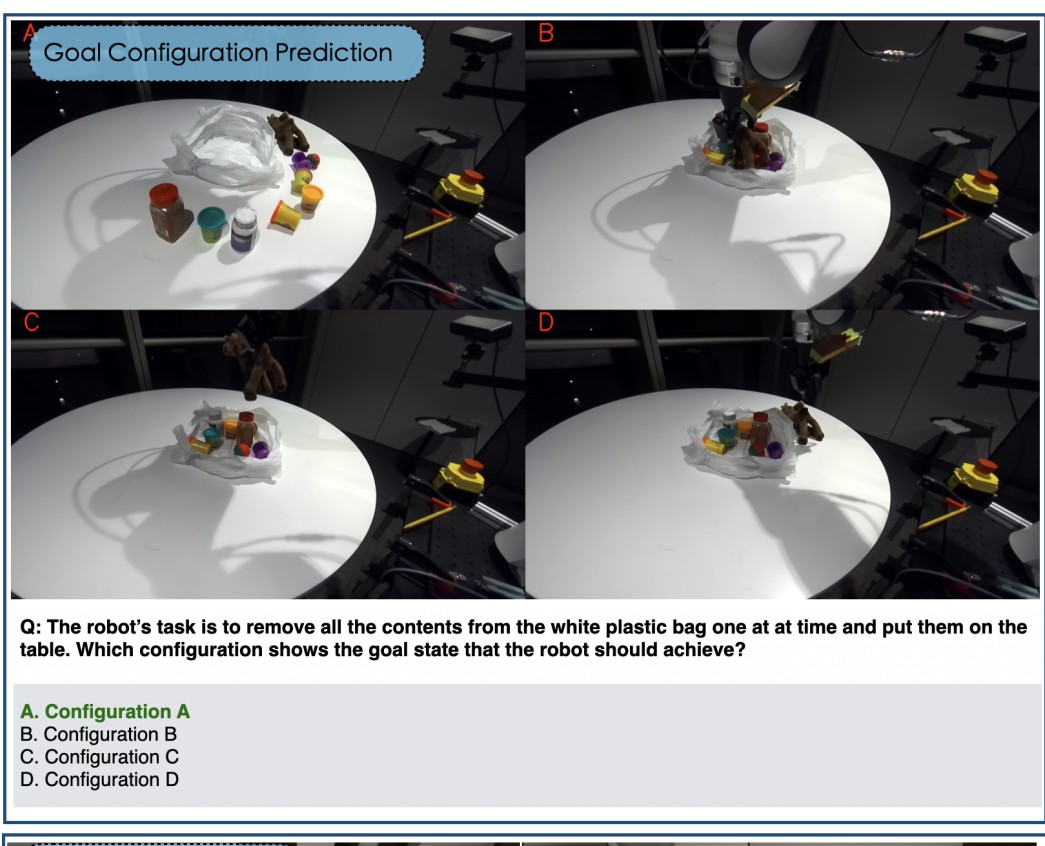

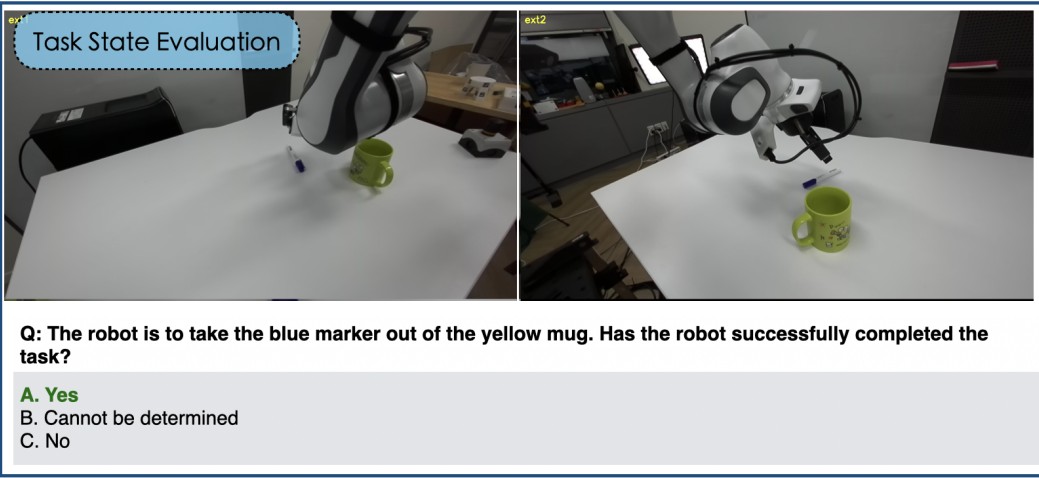

Figure 7: **Example VQAs using with mutlple images.** Each panel presents a unique type of VQA grounded in real-world robot trajectories. Goal Configuration Prediction (top) asks which scene configuration matches the task goal. Task State Evaluation (bottom) queries whether the robot has successfully completed a specified action. These examples demonstrate the need for multimodal reasoning over visual observations and task context. Correct answers are highlighted in green.

## B.2 Challenging Questions

The following figures illustrate several visual question answering (VQA) tasks conducted using robotic trajectories. Each figure presents a unique scenario where human expertise was used to validate the correctness of robotic actions or spatial understanding based on visual inspection. These are questions human experts consider challenging but answered correctly. We will introduce more details for human expert instruction and feedback in Sec. F.

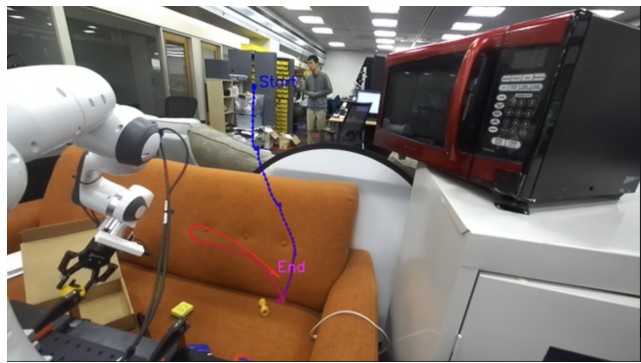

**Question:** Which language instruction best describes the robot's trajectory shown in the image? *[Pick up the black from the drawer, Drop the box into the shelf, put the yellow and black object in the box, Align the black with the table, Move the box to the floor]*

- **Correct Answer:** Put the yellow and black object in the box
- **Expert Rationale:** The trajectory isn't directly at the objects, but the gripper position suggested interaction with the box. This reasoning led me to identify the correct choice clearly.

Figure 8: Identifying the appropriate language instruction corresponding to a robot trajectory involving interaction with a yellow and black object.

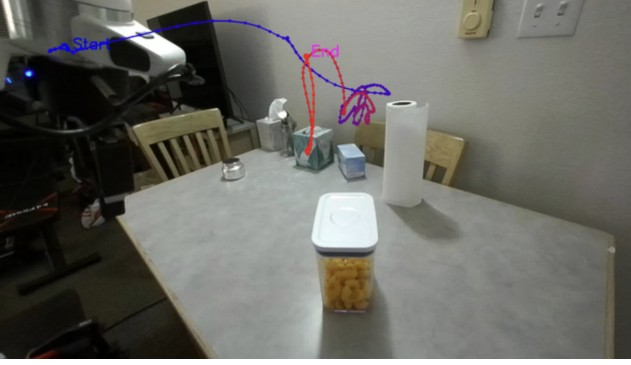

**Question:** Which language instruction best describes the robot's trajectory shown in the image? *[Move the container to the tray, Push the pen towards the bin, Align the box with the drawer, open the container lid, Lift the cup upward]*

- **Correct Answer:** Open the container lid
- **Expert Rationale:** Answers involving absent objects (pen, cup) were quickly eliminated. The trajectory clearly aligned with the container, making the correct answer straightforward.

Figure 9: Robot trajectory clearly aligned with opening a container lid, excluding irrelevant options involving absent items.

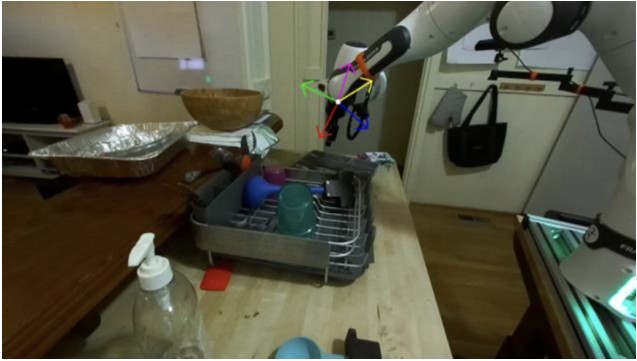

**Question:** The robot task is to move the spoon. Which colored arrow shows the most likely direction the robot will move next?
*[Yellow, Purple, Blue, Green, Red]*

- **Correct Answer:** Red

- **Expert Rationale:** Initially unclear about the spoon's exact position, I carefully inspected to confirm the gripper already grasped the spoon, identifying the red arrow direction correctly.

Figure 10: Discerning the direction of spoon movement based on visual cues, highlighting careful visual analysis.

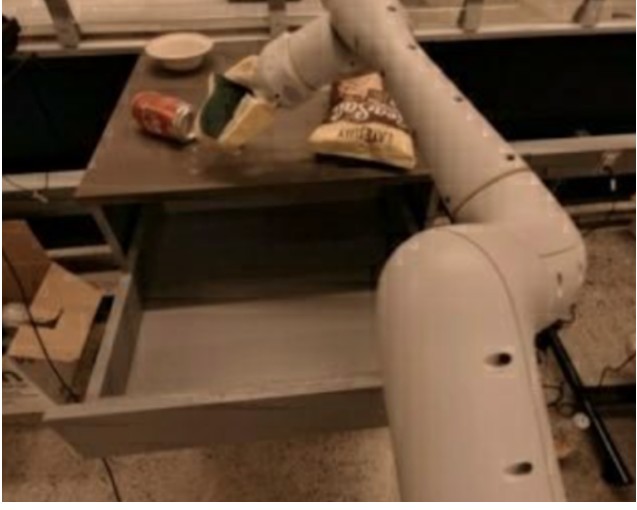

**Question:** Is the robot's grasp of the sponge stable?
*[Yes, No, Cannot be determined, Partially stable]*

- **Correct Answer:** No

- **Expert Rationale:** At first glance, the grip seemed stable, but closer examination revealed the grasp was inadequate on the sponge's edge, confirming instability.

Figure 11: Evaluating the stability of a robotic grasp on a sponge, emphasizing close visual inspection to determine grasp quality.

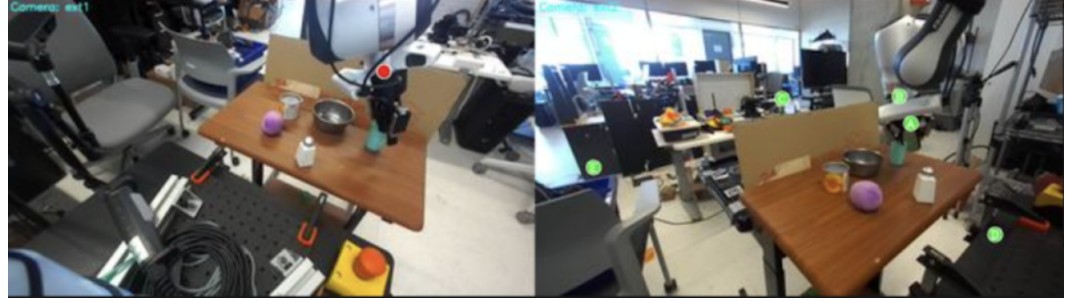

**Question:** In the left image (ext1 camera), a red dot is marked. Which point in the right image (ext2 camera) corresponds closest to this dot?
*[A, B, C, D]*

- **Correct Answer:** D

- **Expert Rationale:** Distinguishing between similarly close points (A and B) required careful analysis. By comparing unique features (such as the wrist camera and the joint's white part), the correct point became evident.

Figure 12: Identifying corresponding points between two camera views, requiring detailed analysis of visual similarities.

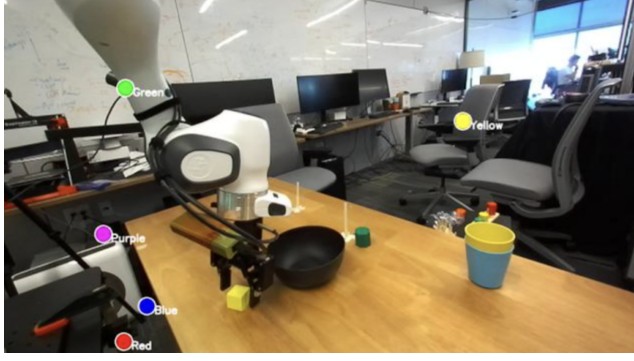

**Question:** In the ext2 camera image, which colored point is closest to the camera?
*[A, B, C, D, E]*

- **Correct Answer:** C

- **Expert Rationale:** Initially, I assumed the blue point was on the table, but a second look revealed it was on the floor, indicating clearly that point C was closest to the camera.

Figure 13: Illustrating depth perception from camera viewpoints and correcting initial visual assumptions upon closer inspection.

# C VLM Output Examples

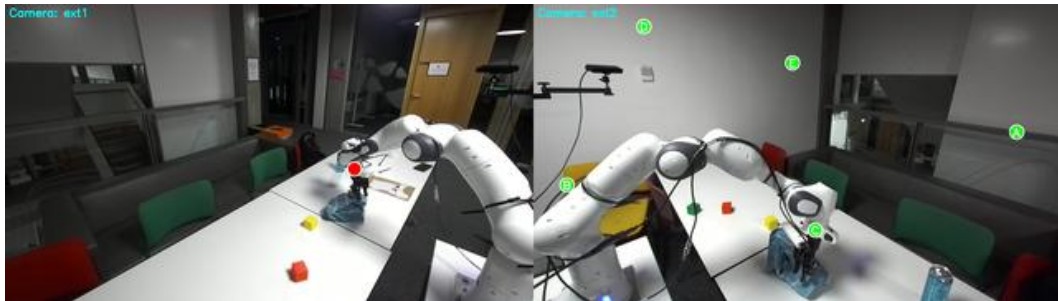

Figure 14: **Question**: In the left image (ext1 camera), a red dot is marked. Which point is the closest point in the right image (ext2 camera) corresponding to the same 3D location? The correct answer is **D. point C**.

---

**Model:** `llava-hf/llava-1.5-7b-hf`

**Prediction:** The closest point in the right image (ext2 camera) corresponding to the same 3D location as the red dot in the left image (ext1 camera) is **point B**.

**Step-by-step explanation:**

- In the left image, the red dot is marked at the top-left corner.
- In the right image, point B is also located in the top-left corner.
- The red dot's position is consistent across both views.
- Thus, point B corresponds to the same 3D location as the red dot.

**Final Answer:** B ✗

---

**Model:** `Qwen/Qwen2.5-VL-7B-Instruct`

**Prediction:** To determine the corresponding point in the right image, we analyze spatial relationships in both views.

**Step-by-step explanation:**

- The red dot in the left image is on the table, slightly left of center and near the bottom.
- In the right image, a small red object appears in the same relative position.
- Candidate analysis:
  - A: Point B — too far from the red object.
  - B: Point A — located on the wall.
  - C: Point E — not aligned spatially.
  - D: Point C — best matches the red dot position.
  - E: Point D — farther than Point C.

**Final Answer:** D ✓

---

**Model:** `meta-llama/Llama-3.2-11B-Vision-Instruct`

**Prediction:** Point **E** in the right image corresponds to the same 3D location as the red dot in the left image.

**Step-by-step explanation:**

- Locate the red dot in the left image (ext1 camera) — it's on the table in front of the robot.
- In the right image (ext2 camera), observe all labeled points.
- Point E is spatially aligned with the red dot's 3D location.

**Final Answer:** E ✗

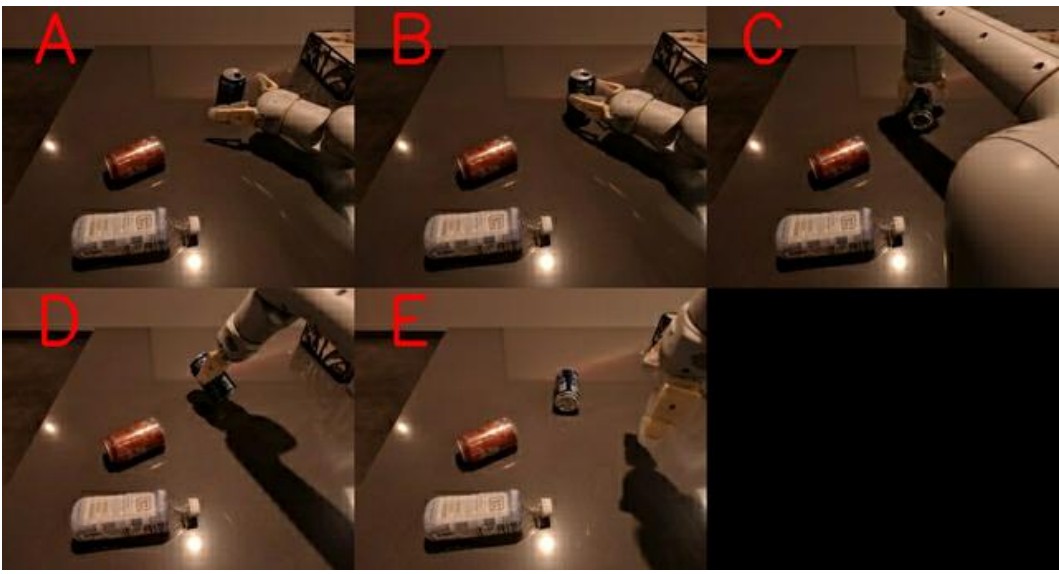

Figure 15: **Question**: The robot's task is to place pepsi can upright. Which configuration shows the goal state that the robot should achieve? [ "Configuration D", "Configuration E", "Configuration B", "Configuration C", "Configuration A" ]

---

**Model:** `llava-hf/llava-v1.6-mistral-7b-hf`

**Prediction:** To determine the configuration that shows the goal state for the robot, we consider:

- The task requires placing Pepsi cans upright.
- The robot starts with cans on their sides.
- We must identify the configuration where the cans are fully upright.

**Analysis of configurations:**

- **Configuration A:** One Pepsi can is upright; two are on their sides.
- **Configuration B:** All cans are on their sides.
- **Configuration C:** One can is upright; two are on their sides.
- **Configuration D:** Two cans upright, two on their sides.
- **Configuration E:** All cans are on their sides.

**Final Answer:** A ✗

---

**Model:** `Qwen/Qwen2.5-VL-7B-Instruct`

**Prediction:** To determine which configuration shows the goal state that the robot should achieve, let's analyze the sequence of images step by step:

- **Configuration A:** The Pepsi can is initially lying on its side.
- **Configuration B:** The can has been moved to a vertical position.
- **Configuration C:** The can remains vertical but is shifted slightly to the right.
- **Configuration D:** The can is still transitioning toward an upright position.
- **Configuration E:** The can is upright and appears stable.

Given the goal of achieving a final, stable upright position:

- B and D are transitional.
- C is upright but not final.
- E is upright and stable.

**Final Answer:** E ✓

---

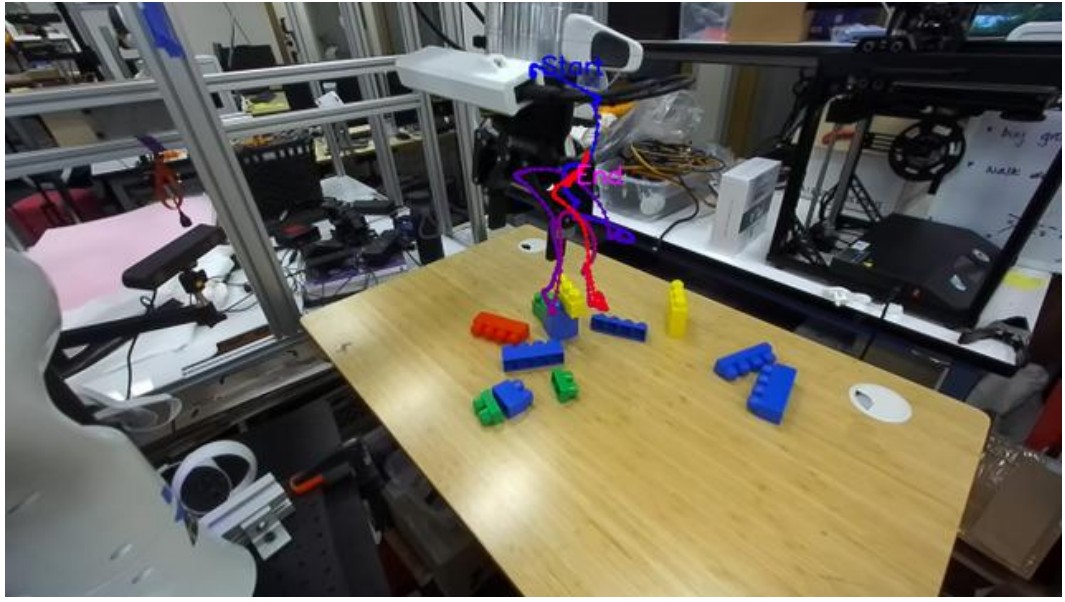

Figure 16: **Question**: The robot's task is to place pepsi can upright. Which configuration shows the goal state that the robot should achieve? [ "Configuration D", "Configuration E", "Configuration B", "Configuration C", "Configuration A" ]

> - C – Incorrectly dismissed, despite covering the multi-step nature of the trajectory.
>
> The model concludes the most likely intent is to grab the block:
> **Final Answer:** B ✗

---

**Model:** `meta-llama/Llama-3.2-11B-Vision-Instruct`

**Prediction:** The robot's trajectory involves multiple steps:

- Rotating the tower clockwise
- Grabbing a block with the gripper
- Picking up the block from the floor
- Dropping the tower into a tray

Each option in the list describes only a partial action:

- A – Rotation only
- B – Grabbing only
- D – Picking up only
- E – Dropping only

Since no single instruction fully describes the trajectory, the model selects: **Answer:** `C. None of the above` ✓

---

# D   Prompt Design for VQA Inference and Verification

This section presents the prompt design strategies used for three key VQA inference settings: answer verification, zero-shot inference, and chain-of-thought (CoT) reasoning. The verifier prompt guides a model to extract and isolate the correct multiple-choice answer from a generated explanation, ensuring alignment between reasoning and final answer format. The zero-shot prompt enforces concise behavior by instructing the model to output only the letter corresponding to the correct answer without additional reasoning. In contrast, the CoT prompt encourages step-by-step reasoning before concluding with the final answer, enabling the model to explain its decision-making process. Additionally, Table 4 outlines prototype question types used in Robo2VLM.

## D.1   Prompt for Verifier

The verifier prompt is used to post-process model-generated answers that contain free-form text, such as in CoT or long-form reasoning outputs. It instructs the model (or another lightweight parser model) to extract the final answer option—typically a letter (A, B, C, D, or E)—from the full response. This prompt plays a critical role in decoupling reasoning quality from answer accuracy, allowing us to evaluate whether the model reaches a correct conclusion after potentially verbose reasoning. The design includes an illustrative example to make the extraction instruction explicit and reduce hallucination of unexpected formats.

---

**Example: Verifier Prompt**

**Instructions:** Please read the example below and extract the final answer from the model response. *Hint:* Your output should be a single letter (e.g., A, B, C, or D) indicating the correct option.

**Question:** What fraction of the shape is blue?
**Choices:** (A) 3/11    (B) 8/11    (C) 6/11    (D) 3/5

**Model response:** The correct answer is (B) 8/11.

**Extracted answer:** B

---

## D.2 Prompt for Zero-Shot

The zero-shot prompt is optimized for direct evaluation of pretrained VLMs without any in-context demonstrations. It instructs the model to select one option from a multiple-choice question using only the corresponding letter. The prompt avoids any reasoning cues or explanations, forcing the model to rely entirely on its pretrained visual and language priors. This prompt setting allows us to assess the model's default grounding and answer formulation capabilities, free from inductive biases introduced by reasoning scaffolds.

---

**Prompt: Zero-Shot Inference**

**Instructions:** Answer the following multiple-choice question by selecting the correct option letter only.
*Hint:* Do not include any explanation—your response should only contain one of the letters: A, B, C, D, or E.
{Question}

---

The {question} can be found in Table 4.

## D.3 Prompt for Chain-of-Thoughts

To improve performance on questions that benefit from intermediate reasoning steps (e.g., spatial inference, task planning, or temporal prediction), we adopt a CoT prompt that encourages step-by-step explanation before committing to a final answer. The CoT prompt explicitly requests both reasoning and a conclusive answer in a standard format, helping the model avoid trailing off or omitting a definitive choice. This setting is particularly useful for analyzing the internal decision-making process of large language models in complex manipulation scenarios.

---

**Prompt: Chain-of-Thought Reasoning**

**Instructions:** Answer the following multiple-choice question by reasoning step by step. Show your work for each step before concluding.
*Hint:* After completing your reasoning, output only the final answer option letter (A, B, C, D, or E) at the end.
{Question}

---

The {question} can be found in Table 4.

Table 4: Question Prompt Templates for VQA Functions

| VQA Function | Question Prompt Prototype |
|---|---|
| `robot_gripper_open` | Is the robot's gripper open? |
| `object_reachable` | Is there any obstacle blocking the robot from reaching {object}? |
| `relative_direction` | In the image from {camera} at step {step}, which direction is the {object} relative to the robot's end effector? |
| `relative_depth` | In the image from {camera}, which colored point is closest/farthest from the camera? |
| `view_correspondence` | In the left image ({camera1}), a red dot is marked. Which point in the right image ({camera2}) corresponds to the same location? |
| `task_success_state` | The robot is to {instruction}. Has the robot successfully completed the task? |
| `is_stable_grasp` | Is the robot's grasp of the {object} stable? |
| `goal_configuration` | The robot's task is to {instruction}. Which configuration shows the goal state? |
| `action_understanding` | The robot is tasked to {instruction}. Which phase of the grasp action is shown? |
| `next_action` | After {current phase}, what will be the robot's NEXT action phase? |
| `trajectory_understanding` | Which language instruction best describes the robot's trajectory shown in the image? |
| `action_direction` | Which colored arrow correctly shows the direction the robot will move next? |
| `temporal_sequence` | What is the correct sequence of action phases shown in the images? |

# E  Fine-Tuning and Evaluation Details

## E.1  Fine-Tuning Details

**Model Configuration** The model utilized for vision-language tasks is based on `meta-llama/Llama-3.2-11B-Vision`, configured for optimal performance. Key settings include gradient checkpointing with the "unsloth" method, a LoRA (Low-Rank Adaptation) rank of 128, an alpha parameter of 256, and no dropout for LoRA modules. Model fine-tuning is selectively enabled for language layers, attention modules, and MLP modules while keeping vision layers fixed. The maximum sequence length is set to 2048 tokens to accommodate complex vision-language interactions.

**Training Setup** Training utilizes the dataset `keplerccc/ManipulationVQA-60k` with a dedicated train split and a validation ratio of 5%. Batch size is carefully controlled at 4 samples per device, enhanced by gradient accumulation over 4 steps. The training process involves linear scheduling of the learning rate, starting at , and includes a weight decay of 0.01. The training is configured to run for one epoch with frequent checkpoints every 1000 steps, evaluation intervals at 5000 steps, and logging every 10 steps.

**Evaluation Protocol** Evaluation is conducted using a maximum of 10,000 test samples, with explicit configuration for generating visualizations and fallback strategies in case of missing test

splits. Generation settings include sampling with a temperature of 0.7 and allowance of up to 50 new tokens per generation. The evaluation setup includes assessing both base and fine-tuned model versions, each clearly delineated within the configuration.

**Distributed Training and Precision** The system leverages distributed training techniques, exploiting high-performance computational resources for scalable training. It utilizes Brain Floating Point (BF16) precision to balance computational efficiency and numerical stability, eschewing FP16 for better performance stability.

## E.2 Evaluation Details

**Experimental Setup** We conducted evaluations using a vision-language model (VLM) pipeline configured specifically for Visual Question Answering (VQA) tasks. The evaluation utilizes the Hugging Face dataset named `keplerccc/ManipulationVQA`, specifically the `test` split, enabling standardized comparisons. To maintain computational efficiency and manage GPU resources effectively, the evaluation employs adaptive batch processing strategies.

**Model Configuration** The evaluation primarily considers two large-scale multimodal models: `llava-hf/llava-v1.6-34b-hf` and `llava-hf/llava-next-72b-hf`. These models leverage tensor parallelism set to 4, harnessing the full computational power of four A100 GPUs to optimize throughput. The models were initialized with a GPU memory utilization parameter set to 0.9, ensuring efficient memory usage without exceeding GPU capacity.

**Prompt and Response Extraction** Each evaluation prompt explicitly instructs the models to select from multiple-choice answers (options A, B, C, D, E). Responses are subsequently processed using a secondary extraction model (`meta-llama/Llama-3.2-3B-Instruct`), designed to deterministically extract the selected letter-answer from the models' verbose outputs. This extraction leverages zero-temperature sampling to guarantee reproducibility and consistency across evaluations.

**Dataset and Evaluation Metrics** The dataset comprises a randomly shuffled subset of test questions, limited by a configurable maximum sample parameter. Accuracy metrics are computed overall and further broken down by tags to provide granular insights into model performance across different question categories. Detailed timing information for responses is recorded to assess computational efficiency, reporting average response times alongside accuracy metrics.

# F Human Expert Instruction and Feedback

To improve the quality and answerability of automatically generated questions, we ask a human expert to improve the data generation process. We provided an initial set of 200 question-image pairs generated by the Robo2VLM pipeline to a human expert for review. The expert was instructed to identify unanswerable or ambiguous cases and annotate the reasons, which were then used to iteratively refine the prompt and generation pipeline. The human expert takes two hours to complete the evaluation. We then follow the revised questions to generate the whole dataset.

## F.1 Evaluation Protocol

The human expert was asked to assess whether each question could be reliably answered based solely on the visual input and accompanying instruction. For cases deemed unanswerable, the expert selected from predefined failure modes including: (1) insufficient or unclear visual context, (2) ambiguous or underspecified language in the prompt, and (3) other task-specific issues. This structured feedback guided the refinement of question templates, robot state annotations, and visual preprocessing steps.

## F.2 Feedback-Driven Refinement of Auto-Curation

Table 5 summarizes the key issues uncovered through human evaluation and the corresponding solutions incorporated into the Robo2VLM pipeline. These challenges fall into four main categories: (i) *Context and Task Definition*, addressing missing goal descriptions and task phase awareness; (ii) *Visual Information and Camera Limitations*, such as limited visibility or poor resolution, which

Table 5: Problems Identified by Human Experts and Corresponding Solutions Implemented in Robo2VLM Pipeline

| Problem Category | Implemented Solution |
|---|---|
| **Context and Task Definition** | |
| Image understanding issue without task context | Enhanced question prompt with task context |
| Lack of goal specificity | Enhanced question prompt with goal descriptions |
| Assumed implicit knowledge of robotic tasks | Added description of the robot's current phase |
| **Visual Information and Camera Limitations** | |
| Limited wrist camera view and object visibility | Integrated multi-view images |
| Invisible gripper state from certain angles | Added gripper state verification and filtering |
| Insufficient image resolution for detailed object identification | Filtered out images with resolution lower than $100 \times 100$ pixels |
| **Question Formulation and Consistency** | |
| Ambiguous or complex question phrasing | Standardized linguistic templates |
| Inconsistent task completion criteria | Unified success state definitions |
| Redundant or confusing phrasing | Applied phrase filtering and clarity scoring |
| Conflicting answers across questions for same image | Added consistency validation checks |
| **Category-Specific Issues** | |
| Multiple viewpoints needed for configuration selection | Added multi-angle verification to configuration questions |
| Spatial reasoning depends on object boundaries and color | Improved spatial questions with object detection and color validation |
| Direction prediction depends on task goal | Integrated goal-aware motion prediction |

were mitigated through multi-view integration and filtering heuristics; (iii) *Question Formulation and Consistency*, where we standardized linguistic structures, unified success criteria, and added consistency validation checks; and (iv) *Category-Specific Issues*, including configuration reasoning, spatial alignment, and directional prediction, which were resolved using goal-aware and multi-perspective analysis. Together, these improvements enhance the reliability, interpretability, and generalization of vision-language evaluations in robotic settings.

# G  Key dataset statistics

We analyzed a total of 60,000 samples in the dataset. On average, questions are 108.69 characters long, with a median length of 113 characters. The shortest question contains 28 characters, while the longest reaches 378 characters. Each question includes an average of 4.65 answer choices, with most having either 4 or 5 options. The typical choice is 14.22 characters long on average, though lengths vary widely—from as short as 1 character to as long as 271 characters. The combined length of all choices per question averages 66.09 characters, with a median of 44 characters and a range from 5 to 687 characters.

In terms of correct answer distribution, the dataset is relatively balanced among options A to D: 22.03% of correct answers are 'D', 21.86% are 'B', 21.74% are 'C', and 21.53% are 'A'. Option 'E' appears less frequently, making up 12.84% of correct responses.

Regarding image data, the average image width is 520.66 pixels, with a median of 640 pixels, while heights average 292.99 pixels, with a median of 256 pixels. Image widths range from 84 to 640 pixels, and heights from 84 to 480 pixels. The most common image resolutions are 640x360 (39.61%), 320x256 (21.14%), 640x240 (8.46%), 640x180 (5.81%), and 448x224 (4.14%). Across the dataset, there are 19 unique image resolutions.

Table 6: Dataset Statistics Summary for 60,000 Samples

| Category | Metric | Value |
|---|---|---|
| Questions | Avg. length (chars) | 108.69 |
| | Median length (chars) | 113.00 |
| | Min length (chars) | 28 |
| | Max length (chars) | 378 |
| Choices | Avg. # choices per question | 4.65 |
| | Median # choices | 5.00 |
| | Min # choices | 4 |
| | Max # choices | 5 |
| | Avg. length of a choice (chars) | 14.22 |
| | Median length of a choice (chars) | 6.00 |
| | Min/Max choice length (chars) | 1 / 271 |
| Choices (total per question) | Avg. total length (chars) | 66.09 |
| | Median total length (chars) | 44.00 |
| | Min/Max total length (chars) | 5 / 687 |
| Answer Distribution | A | 12,918 (21.53%) |
| | B | 13,115 (21.86%) |
| | C | 13,046 (21.74%) |
| | D | 13,216 (22.03%) |
| | E | 7,705 (12.84%) |
| Image Width (px) | Avg. | 520.66 |
| | Median | 640.00 |
| | Min/Max | 84 / 640 |
| Image Height (px) | Avg. | 292.99 |
| | Median | 256.00 |
| | Min/Max | 84 / 480 |
| Top-5 Resolutions | 640x360 | 23,768 (39.61%) |
| | 320x256 | 12,683 (21.14%) |
| | 640x240 | 5,075 (8.46%) |
| | 640x180 | 3,484 (5.81%) |
| | 448x224 | 2,482 (4.14%) |
| | Unique resolutions | 19 |

## H Distractor Choice Design

This section outlines the design and evaluation of distractor choices in our VQA dataset, which play a critical role in determining question difficulty and diagnostic value. We begin by examining the impact of introducing a "None of the Above" (NAB%) option, which systematically increases task ambiguity and reduces model performance across the board (Fig. 17). We then detail the principles and heuristics used to generate diverse and context-aware distractors for different question types. These include binary negations, categorical sampling, spatial reasoning perturbations, and content-aware language distractors. Special emphasis is placed on generating plausible incorrect choices that reflect partial knowledge, ambiguity, or visually confusable elements. Finally, we describe how randomized shuffling and probabilistic replacement with NAB options further strengthen the challenge by discouraging rote pattern matching. Together, these strategies enhance the dataset's ability to probe fine-grained reasoning, visual grounding, and robustness to uncertainty in large vision-language models.

## H.1 None of the Above Proportion

This section shows experiment of adding 'None of the Above' selection Ratio (NAB%) choice increase the difficulty of the dataset and model accuracy decrease for all the models. We show the result in the line plot in Fig. 17.

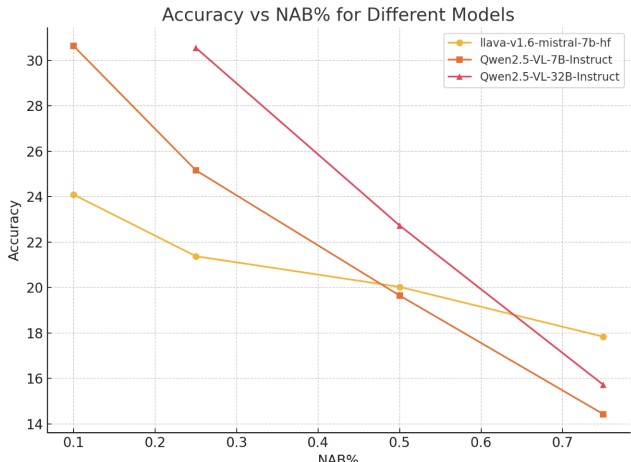

Figure 17: Accuracy vs. 'None of the Above' Selection Ratio (NAB%) for Three Vision-Language Models

The plot reveals that all three models experience a decline in accuracy as NAB% increases, indicating reduced confidence or higher prediction difficulty when a greater proportion of questions are considered potentially unanswerable. Qwen2.5-VL-32B-Instruct consistently outperforms the other two models when data is available, achieving the highest accuracy of 30.55% at NAB% = 0.25. Interestingly, the 7B Qwen2.5-VL variant initially performs well (30.63% at NAB% = 0.1) but degrades more sharply than the 32B version. The llava-v1.6-mistral-7b-hf model maintains the lowest accuracy across all NAB% levels, suggesting it is less robust under ambiguity. These trends highlight the importance of model scale and training data in handling tasks with varying uncertainty.

## H.2 Distractors

The design of distractor choices is crucial for creating challenging and meaningful Visual Question Answering (VQA) instances. The provided Python codebase employs several strategies to generate plausible yet incorrect options, aiming to test nuanced understanding rather than simple pattern recognition.

**Binary and Generic Distractors** For questions anticipating a binary response (e.g., Yes/No), the primary distractor is often the direct negation of the correct answer. This is evident in functions like `vqa_robot_gripper_open` and `vqa_object_reachable`. These are typically supplemented by generic distractors such as "Cannot be determined" or context-specific but still general alternatives like "Partially open" or "Partially reachable". The `_validate` method ensures that binary questions have exactly four choices, accommodating these patterns.

**Categorical and Permutation-Based Distractors** Many VQA generation functions define a set of possible categories and select distractors from those not matching the correct answer. **Relative Directions:** In `vqa_relative_direction`, a comprehensive list of possible spatial relations (e.g., "Upper Left", "Lower Forward") is generated. After identifying the correct direction, incorrect choices are drawn from this list, with a preference for those sharing some component (e.g., the same vertical component) with the correct answer to increase plausibility. **Action Phases:** For `vqa_action_understanding` and `vqa_next_action`, distractors are chosen from a defined set of robot action phase descriptions (e.g., "Approaching the object with open gripper", "Firmly grasping the object"). The incorrect choices are the descriptions of other valid phases. **Temporal Sequences:** `vqa_temporal_sequence` generates distractors by creating incorrect orderings (permutations) of

the actual sequence of events or phases if the question is about the sequence itself. **Color/Label-Based Choices:** In `vqa_relative_depth` and `generate_action_direction_selection_vqa`, distinct colors (e.g., "Red", "Green", "Blue") are assigned to different points or arrows in the image. The choices are then these color names, with one corresponding to the correct visual marker. Similarly, `vqa_multi_view_correspondence` uses letter labels ("A", "B", "C", "D", "E") for choices corresponding to marked points.

**Spatially Derived Distractors**  For tasks involving spatial reasoning, distractors are often generated to be distinct in the image space. In `vqa_multi_view_correspondence`, distractor points are generated in different quadrants of the image from the correct corresponding point, ensuring a minimum pixel distance from each other and the correct point. `generate_action_direction_selection_vqa` creates incorrect directional arrows by ensuring their angles are meaningfully different from the correct action direction, with a minimum angular separation.

**Content-Based Distractors from External Knowledge**  The `vqa_trajectory_understanding` function generates distractor language instructions by using templates (e.g., "Pick up the {} from the {}") and filling them with common objects and locations, which may or may not be present in the current scene, thus testing a deeper understanding of the visualized trajectory against plausible alternative tasks.

**Strategic Shuffling and "None of the above"**  The `_shuffle_choices` method is systematically called after initial VQA construction. This method randomizes the order of the correct answer and the initially formulated incorrect choices. Furthermore, for non-binary questions (typically those with five choices), there is a 20% chance to replace the actual correct answer with "None of the above", and the original correct answer text is then discarded for that instance, making "None of the above" the correct choice. This adds another layer of complexity, requiring the system to not only identify the correct option but also to recognize when none of the substantive options are correct.

The combination of these strategies ensures a diverse set of distractors, tailored to the specific type of question being posed and the visual information presented.

# I   Additional Experimental Results

## I.1   Blind VLM Baseline

We conducted an ablation study under the "Blind" VLM setting, where visual inputs were removed and only text was provided. For 1,000 randomly sampled questions from the Robo2VLM-1 dataset, the random baseline accuracy was $21.5\%$. The results show that without vision, model performance drops significantly.

Table 7: Comparison between Blind and Standard VLMs on Robo2VLM-1 (1000 samples).

| Model | Blind LLM (%) | Standard VLM (%) |
|---|---|---|
| GPT-4o-mini | 16.0 | 29.7 |
| ChatGPT-4o | 21.7 | 41.5 |
| LLaVA 1.5-7B | 22.4 | 22.6 |
| Qwen-7B | 24.3 | 34.8 |

## I.2   Closed-Source Model Evaluation

We evaluated closed-source models under the same setup as open-source baselines. GPT-4o shows competitive results comparable to the strongest open models. We note that all the models are released at different time.

Table 8: Closed- and open-source VLM comparison on Robo2VLM-1.

| Model | RS | OS | SR | SU | MV | TS-G | TS-S | TS-GL | AU | IP | TU | Overall |
|-------|-----|-----|-----|-----|-----|------|------|-------|-----|-----|-----|---------|
| GPT-4o-mini | 12.0 | 74.2 | 20.6 | 15.9 | 21.4 | 7.3 | 29.8 | 13.8 | 32.4 | 68.8 | 58.3 | 29.7 |
| GPT-4o (2024-08-06) | 42.7 | 93.3 | 17.0 | 35.3 | 31.0 | 14.6 | 44.9 | 16.5 | 33.8 | 83.8 | 70.8 | 41.5 |
| Qwen2.5-VL-32B | 48.9 | 90.5 | 18.8 | 29.2 | 19.8 | 35.2 | 60.4 | 18.7 | 32.2 | 71.4 | 49.3 | 41.3 |

## I.3 Effect of Paraphrased Questions on Model Performance

We used Qwen 3-8B to paraphrase 1,000 template-based questions and answers to introduce langauge variability. Model accuracy remained stable, which suggests that Robo2VLM evaluation is robust to moderate linguistic variations.

Table 9: Effect of paraphrased question variants on VLM accuracy (%).

| Model | Original | Paraphrased |
|-------|----------|-------------|
| LLaVA 1.5-7B | 22.6 | 21.5 |
| LLaVA 1.6-7B | 23.4 | 23.2 |
| Qwen2.5-VL-7B | 34.8 | 34.0 |

