# OpenReview forum: "Robo2VLM: Improving Visual Question Answering using Large-Scale Robot Manipulation Data"
_NeurIPS.cc/2025/Datasets_and_Benchmarks_Track — NeurIPS 2025 Datasets and Benchmarks Track spotlight_

### Official Review · Reviewer_r1zP · 2025-06-19

**Rating:** 5
**Confidence:** 3

**Summary:**

The paper introduces **Robo2VLM**, a framework and dataset for sensor-grounded Visual Question Answering (VQA) in robotics. Key contributions include: 1. Robo2VLM Framework 2. Robo2VLM-1 Dataset 3. Diverse VQA Types 4. Model Evaluation
In summary, Robo2VLM enables scalable VQA dataset generation for robotics and enhances VLM evaluation on complex physical reasoning tasks.

**Additional Feedback:**

Including qualitative examples of typical VLM errors (e.g., incorrect spatial reasoning or misidentified actions) would help readers better understand current model limitations and dataset challenges.

Could Robo2VLM be extended to support open-ended or free-form VQA, beyond multiple-choice format?

**Dataset Code Accessibility:**

Yes

**Dataset Code Comments:**

The dataset and code is publicly available.

**Ethical Comments:**

The dataset is generated from pre-existing robot manipulation trajectories (Open X-Embodiment), and no human subjects or sensitive data are involved.

**Ethical Considerations:**

No, there are no or only very minor ethics concerns

**Final Justification:**

I have no futher question.

**Limitations Weaknesses:**

1. The framework relies on the quality and calibration of teleoperated robot trajectories. As acknowledged in Sec. 6, miscalibrated or incomplete sensory data (e.g., NYU VINN dataset) can reduce the quality and quantity of generated VQA items.

2.   Although the dataset is large, question templates are hand-designed, which may limit linguistic variety and introduce unintentional biases.

3. The paper lacks demonstration of how models trained on Robo2VLM perform in real robot deployment settings.

**Strengths Contributions:**

The paper is **well-written, structured**, and flows logically from motivation to method, dataset, experiments, and results.

Significance & Novelty
- Introduces Robo2VLM, a novel framework that generates VQA from real robot teleoperation data using multi-modal sensory inputs—addressing the lack of spatial and interaction grounding in existing VLM benchmarks.
- Presents Robo2VLM-1, offering unprecedented diversity across scenes, tasks, and skills (Sec. 4).

Impact & Relevance
- Enables scalable and automated VQA dataset creation without costly manual annotations.
- Demonstrates that fine-tuning VLMs on Robo2VLM-1 significantly improves spatial and interaction reasoning.

Relation to Prior Work
- Clearly differentiates from simulation-based datasets (e.g., ALFRED, EQA-MX) and manually annotated benchmarks like RoboVQA by leveraging robot proprioception and kinematics for grounded QA generation.
- Builds upon and extends the Open X-Embodiment dataset, showcasing how rich robot data can serve as supervision for language models.

---

> ### Author Rebuttal · Authors · 2025-07-31
>
> Thank you for your positive review and valuable comments. We are glad you like our topic, novelty, and related work\! To your feedback,
>
> * **Miscalibrated or incomplete sensory data reduce quality as acknowledged in Sec 6**
>
> This is a good point that we should clarify in the revision. There are learned methods to augment miscalibrated or incomplete sensory data. For example, in the DROID [1] dataset, the authors patch the miscalibrated robot trajectories through a series of quality assessment and transformations. The augmented data can still be useful by grounding Robo2VLM with well-defined spatial algorithms.
>
> *  **Hand-designed question templates may limit linguistic diversity and introduce bias**
>
> Thank you for the insightful comment\! We agree that limited linguistic diversity may introduce unintended biases.
>
> To enhance the linguistic diversity, a common practice in the VQA literature is to ask human annotators to rewrite questions while preserving their semantics. Recent work has increasingly adopted LLMs for automated paraphrasing to achieve broader linguistic coverage [2]. As a preliminary exploration, we hypothesize that, paraphrasing with LLM would not lose semantic information, because our hand-designed question templates are linguistically simple and VQA diversity comes from vision.
>
> To test the hypothesis, we use Qwen 3-8B to paraphrase 1000 template-based questions and answers in Robo2VLM. As shown in the table below, model performance remains consistent under paraphrased inputs, suggesting that Robo2VLM can support more diverse question formulations without compromising evaluation stability.
>
> |  | Robo2VLM  | Parapharsed Robo2VLM |
> | :---- | :---- | :---- |
> | LLaVA1.5 \- 7B | 22.6% | 21.5% |
> | LLaVA 1.6 \- 7B | 23.4% | 23.2% |
> | Qwen 2.5 VL \- 7B | 34.8% | 34.0% |
>
> In the revised draft, we plan to further investigate how linguistic variation affects model performance and bias in VQA generation of Robo2VLM.
>
> * **Lack demonstration of how models trained in real robot deployment setting**
>
> We agree that bridging benchmark performance to real-world deployment is an important direction, and we plan to explore this in future work. Robo2VLM is designed primarily as a data generation and benchmarking framework for VLMs. We are integrating models trained by Robo2VLM to realistic robot settings at robotics training data curation and quality control. We will provide more details in Robo2VLM real world use cases and demonstration in the revised version.
>
> * **Could Robo2VLM be extended to support open-ended or free-form VQA beyond multiple-choice format?**
>
> This is a great idea\! Robo2VLM could potentially be extended to support open-ended VQA, and we view this as a promising yet challenging direction.  One of the major contributions of Robo2VLM is that we carefully design unambiguous answers and distractors with procedural algorithms based on robotics domain knowledge and grounded in multimodal sensory data. If using open questions for benchmarking in the current Robo2VLM,  it is hard to match to a known ground truth such as the trajectory understanding and other goal-conditioned reasoning. In the future work, we will explore open-ended questions with human evaluation and LLM-as-a-judge for open-ended answers.
>
> * **Include examples of typical VLM errors such as incorrect spatial reasoning or misidentified actions**
>
> This is a great suggestion\! We agree that showcasing typical VLM failure modes (e.g., incorrect spatial reasoning or misidentified actions) is important for illustrating model limitations and dataset challenges. We currently include such examples in Appendix B.2, and we will make them more clear to the reader by linking them directly in the main experiment section for improved visibility and clarity.
>
> references:
> [1] Khazatsky, Alexander, et al. "Droid: A large-scale in-the-wild robot manipulation dataset." arXiv preprint arXiv:2403.12945 (2024).
> [2] Yadav, V., Tang, Z., & Srinivasan, V. (2024, July). Pag-llm: Paraphrase and aggregate with large language models for minimizing intent classification errors. In Proceedings of the 47th international ACM SIGIR conference on research and development in information retrieval (pp. 2569-2573).

---

### Official Review · Reviewer_Bctk · 2025-07-01

**Rating:** 5
**Confidence:** 4

**Summary:**

This paper introduces Robo2VLM, a Visual Question Answering (VQA) dataset generation framework. Compared to previous embodied VQA datasets, Robo2VLM extends the scope by incorporating additional non-visual and non-descriptive sensory modalities, such as end-effector pose, gripper aperture, and force sensing. This makes VQA more comprehensive and better aligned with robotics. The paper systematically describes the dataset construction process and evaluates multiple downstream VLM models.

**Additional Feedback:**

I encourage the authors to carefully consider the weaknesses raised. I look forward to the authors' responses. If my comments involve any misinterpretation, I would appreciate further discussion and clarification. Thank you.

**Dataset Code Accessibility:**

Yes

**Dataset Code Comments:**

The authors have provided complete open-source access, including the dataset generation process and the public release of the dataset within the open-source community.

**Ethical Considerations:**

No, there are no or only very minor ethics concerns

**Final Justification:**

The authors greatly addressed my main concerns, including the method of instruction parsing and using closed-source VLM for more validation. Therefore, I will maintain my overall positive score and improve my confidence score.

**Limitations Weaknesses:**

- The Robo2VLM framework relies on several foundation models for assistance, such as using Qwen2.5 for instruction parsing. How can we ensure that this process is accurate? Is there any error detection mechanism in place to prevent the generation of erroneous data?

- In the downstream VLM experiments, the baselines being compared are primarily open-source models, which may not provide a comprehensive evaluation of the performance of different VLMs on the dataset. Would it be possible to incorporate some closed-source large models (e.g., GPT-4, Kimi, Gemini) for a more systematic evaluation of robot VQA?

**Strengths Contributions:**

- The paper is well-written and easy to read.

- The data generation process for Robo2VLM is based on a coherent sequence of operations, rather than randomly selecting pairs from the dataset. This ensures continuity in the data generation process and contributes to multiple reasoning approaches for VQA. In addition, Robo2VLM extends the scope by incorporating additional robot modalities, e.g., end-effector pose.

- The dataset statistics and downstream model evaluations are well-executed and thorough.

---

> ### Author Rebuttal · Authors · 2025-07-31
>
> Thank you for your thoughtful and constructive review. We sincerely appreciate your recognition of the paper’s strengths and your helpful feedback on areas for improvement. Below we address the specific concerns raised:
>
> * **The Robo2VLM framework relies on several foundation models for assistance, such as using Qwen2.5 for instruction parsing. How can we ensure that this process is accurate? Is there any error detection mechanism in place to prevent the generation of erroneous data?**
>
> Thanks for pointing it out\!  We will clarify in the final draft on how we minimize the usage of the foundation models to ensure accuracy. Foundation models are not on the critical path of Robo2VLM data generation. The dataset is generated using procedural algorithms, as introduced in the section 3, based on robotics domain knowledge and grounded in multimodal sensory data, including end-effector pose, gripper state, and force-torque readings. This process does not use any foundation model.
>
> We agree that using Qwen2.5 for instruction parsing may have inaccuracies. It is still a hot research topic to extract answers from VLM/LLM outputs. We acknowledge that ensuring full accuracy is not possible. xFinder[1] at ICLR 2025 shows that both regular expression and state-of-the-art LLMs face different levels of extraction error. In Robo2VLM, the answer extraction is relatively simple compared to xFinder testing dataset. As shown in Appendix B.2, the LLM needs to convert model output  “The final answer is A” or “\box{A}” to a standardized format like “A”, ensuring consistent and reliable evaluation. In the current evaluation, we combined regular expressions with LLM. We will acknowledge this limitation and clarify in the final version.
>
>
> * **In the downstream VLM experiments, the baselines being compared are primarily open-source models, which may not provide a comprehensive evaluation of the performance of different VLMs on the dataset. Would it be possible to incorporate some closed-source large models (e.g., GPT-4, Kimi, Gemini) for a more systematic evaluation of robot VQA?**
>
> Thank you for the suggestion\!  We will include more closed-source models into the final version. In our initial experiments, we compare GPT-4o and GPT-4o-mini with the same setup as other VLMs in VLM Chain-of-Thought settings for VQA accuracy. To mitigate concerns about data contamination, especially since Robo2VLM-1 has been publicly available and downloaded thousands of times monthly, we used the most recent versions of these models that were released after the dataset’s initial release. Specifically, we report results using GPT-4o (gpt-4o-2024-08-06) and GPT-4o-mini (gpt-4o-mini-2024-07-18), which are the most up-to-date available at the time of submission.
>
> | Model | overall | RS | OS | SR | SU | MV | TS-G | TS-S | TS-GL | AU | IP | TU |
> | ----- | ----- | ----- | ----- | ----- | ----- | ----- | ----- | ----- | ----- | ----- | ----- | ----- |
> | **gpt-4o-mini** | 29.7 | 12.00 | 74.16 | **20.57** | 15.88 | 21.43 | 7.32 | 29.75 | 13.76 | 32.39 | 68.75 | 58.33 |
> | **gpt-4o-2024-08-06** | **41.5** | 42.67 | **93.26** | 17.02 | **35.29** | **30.95** | 14.63 | 44.94 | 16.51 | **33.80** | **83.75** | **70.83** |
> | **Qwen 2.5-VL 32B** | 41.30 | **48.85** | 90.50 | 18.82 | 29.19 | 19.82 | **35.17** | **60.43** | **18.71** | 32.21 | 71.35 | 49.32 |
>
> Compared with the Qwen 2.5-VL 32B released in April 2025, which is the best model in our open source model evaluation, GPT 4o release in 2024 shows a comparable result. We will incorporate the results into the final draft.
>
> [1] Yu, Q., Zheng, Z., Song, S., Xiong, F., Tang, B., & Chen, D. xFinder: Large Language Models as Automated Evaluators for Reliable Evaluation. ICLR 2025

---

> > ### Comment · Reviewer_Bctk · 2025-08-05
> > **Response to the Rebuttal**
> >
> > Dear authors,
> >
> > Thank you so much for your rebuttal! I have carefully read all the comments and the rebuttal. And I'm satisfied with the rebuttal that solves my main concerns, including the method of instruction parsing and using closed-source VLM for more validation. Considering that I've already rated the paper with a positive score, I will further improve the confidence score of this paper.
> >
> > Hope all the useful results during the rebuttal process can be supplemented to the final version.
> >
> > Best,
> >
> > Reviewer Bctk

---

### Official Review · Reviewer_fh7u · 2025-07-01

**Rating:** 5
**Confidence:** 4

**Summary:**

In this work, the authors proposed an inverse paradigm that utilizes tele-operated robot trajectories from Open X-Embodiment to enhance and evaluate VLMs’ capabilities on robotic manipulation-specific tasks. Specifically, they proposed a VQA sample generation framework from real-world robot trajectories named Robo2VLM and a correspondingly curated large-scale VQA dataset, named Robo2VLM-1. To generate VQA samples, Robo2VLM first performs scene-interaction understanding to extract keyframes. Then, based on the pre-defined question prototypes, Robo2VLM performs question conversion and spatial query projection via a visual-language grounding module. The question prototypes can be categorized into Spatial Reasoning, Goal-conditioned Reasoning, and Interaction Reasoning, which are all essential capabilities for the success of a robot agent’s task execution. Multiple current state-of-the-art VLMs have been evaluated on the benchmark established with the Robo2VLM-1 dataset in zero-shot, finetuned, and CoT prompting ways, and detailed analyses are provided in the experiment part.

**Dataset Code Accessibility:**

Partly

**Dataset Code Comments:**

The Huggingface dataset is complete and well-organized. However, the GitHub code seems to be incomplete and left with several placeholders. But overall, the dataset code is clearly open-sourced.

**Ethical Considerations:**

No, there are no or only very minor ethics concerns

**Final Justification:**

This work provides a way to reflect VLM's genuine capabilities to deal with real-world spatial and interaction-oriented understandings in the form of multiple-choice QA. It is of great application value, and the experiments are all-rounded. Therefore, I am prone to give a positive score. However, as put forward by Reviewer FBtn and me, the multiple-choice QA, instead of an open-question form, hinders it from getting a higher score.

**Limitations Weaknesses:**

- In the Robot State task, VLMs are required to estimate gripper/arm position. However, in question prototypes, no spatial priors are provided. It remains doubtful if it is a good way to reflect VLM's spatial understanding capabilities.
- In fact, the questions do not require the VLMs to give the accurate values, but only need to choose from the pre-defined multiple choices.
- In that case, models’ performances may be, to a certain degree, affected by the distribution of the choices and may introduce human priors and preferences.
- How the keyframes are extracted from the robot trajectories is not elaborated? Are they manually selected by human beings?

**Strengths Contributions:**

- Robo2VLM and Robo2VLM-1 provide a way to genuinely and numerically reflect VLMs’ capabilities when dealing with real-world spatial or interaction-oriented understandings, which is of vital importance when evaluating the feasibility of applying VLMs to real-world applications.
- The way that Robo2VLM utilizes multi-modality information provided by the collected robot data to decompose a robot trajectory makes sense and is very efficient and reliable.
- The carefully designed question prototypes and related tasks are all essential for robotic manipulation. Besides, the data distribution of Robo2VLM-1 is diverse enough to evaluate various facets of capabilities that are entailed for reliable executions.
- Extensive experiments have been performed to reflect various VLMs’ capabilities. Further comprehensive analysis of the effects of finetuning with Robo2VLM-1 training data, CoT prompting, and model size is given.

---

> ### Author Rebuttal · Authors · 2025-07-31
>
> Thank you for your very positive review and helpful comments\! We appreciate your recognition of Robo2VLM’s effectiveness in using real robot trajectories to evaluate VLMs on spatial and interaction reasoning. We also value your thoughtful feedback regarding spatial priors and the question design, which we address in detail below.
>
> * **In the Robot State task, VLMs are required to estimate gripper/arm position. However, in question prototypes, no spatial priors are provided.**
>
> Thank you for this insightful comment regarding spatial priors in the Robot State task. While explicit spatial priors are not provided in the question templates, the Robo2VLM dataset implicitly offers strong spatial context. Specifically, the dataset primarily consists of single-arm, tabletop manipulation trajectories using common robot types. As a result, the visual appearance of the arm and gripper is relatively consistent, allowing VLMs to leverage this as a spatial prior. Additionally, because the trajectories are collected from real-world environments, such as offices and labs (see Figure 3), the background scenes are also consistent and contribute additional contextual cues. We will clarify this point in the revised version of the paper.
>
> * **The dataset does not require VLM to give accurate values**
>
> This is a great point! We agree that requiring VLMs to predict precise numeric values (e.g., pixel coordinates or distances) is important yet remains a significant challenge for current models. In addition, judging admissible numeric values is challenging. Therefore, our dataset is designed around multiple-choice spatial and interaction reasoning tasks. We carefully design the correct answer and unambiguous distractors, and emphasize relative understanding rather than absolute measurement. We consider it a future work to bring the same level of unambiguity to   evaluate the spatial and numerical capabilities of VLM.
>
> * **In that case, models’ performances may be, to a certain degree, affected by the distribution of the choices and may introduce human priors and preferences.**
>
> Thank you for raising this subtle but important issue. We agree that the multiple-choice format may introduce biases due to the distribution and positioning of answer choices, which could potentially influence model predictions by reinforcing unintended human priors. To solve this issue, we design distractors using heuristics such as spatial plausibility and visual contrast (Sec. 3.2), we randomly shuffle the choices (Appendix E.2), and we make sure the choices are evenly distributed across all possible choices (Appendix G).
>
> * **How key frames are extracted**
>
> Thanks for pointing out the clarity issue\! This is one of our major contributions. The data Robo2VLM generation pipeline can generate thousands of questions in a one-minute robot trajectory, so selecting questions / frames that are distinct from other frames is important. We partition the robot trajectory into different manipulation phases following the robotics community standard [1] and multi-modal sensory data. A manipulation phase contains a short-horizon unitary action, such as reaching, in contact, moving away. We extract key frames from each manipulation phase. Since frames within a single manipulation phase tend to be semantically similar, we can select the middle frame of each phase as our representative frame. This maximizes semantic distance between selected frames across different phases. We will clarify this in the final draft.
>
> * **Github Code is incomplete**
>
> Thank you for checking out our dataset and code\! We have fully released our data and code. We sealed the code at the NeurIPS branch for review. The documentation has been completed at our current main branch.
>
> [1]  Henrik I. Christensen and Gregory D. Hager. Sensing and estimation. In Bruno Siciliano and Oussama Khatib, editors, Springer Handbook of Robotics, Springer Handbooks, pages 91–112. Springer, 2016
> [2] Makoto Kaneko. Manipulation and Interfaces. Springer Handbook of Robotics, Springer Handbooks, pages 893–1126. Springer, 2016

---

> > ### Comment · Reviewer_fh7u · 2025-08-05
> >
> > Thanks for the authors' rebuttal and further clarification. My concerns are well addressed. Since I've already given a positive score, I am prone to maintain my previous score. Expect to see the final version with all problems and misunderstandings solved.

---

### Official Review · Reviewer_FBtn · 2025-07-02

**Rating:** 5
**Confidence:** 4

**Summary:**

This work presents ROBO2VLM, a novel framework to generate “Visual Question Answering” (VAQ) data from multimodal recordings or tele-operated robot trajectories in the context of robot manipulation tasks.
The motivation is to facilitate the obtention of challenging data to improve existing Vision-Language Models (VLMs) in some of their known weak points, namely spatial and interaction reasoning.
The framework is demonstrated by using it to generate and release a large VAQ dataset (ROBO2VLM-1) consisting of multiple-choice questions where, given an image, the question requests information that needs spatial or interaction understanding. This dataset is evaluated with state-of-the-art open-source VLMs, highlighting the challenge that this kind of data poses for existing models and demonstrating how fine-tuning on the built data improves the performance of one of these models (LlaVA) on most of the tasks.

**Additional Feedback:**

Some small fixes or clarifications that could be added:

- Are the fine-tuned models being released? This doesn’t hinder comparisons but it would be useful and facilitate replicability.

- Description of table 2 mentions that detailed data modalities is included in that table, but just before the text mentions data calibration, which is not included in the table. It would be more clear to point to a table where all modalities included from each dataset would be compiled. What are those extra modalities recovered for some? (“manually include modalities introduced by the original paper”, in lines 191 - 192)

- End of section 4 could be a bit more explicit about how the curation process is done. By who? External annotators? Authors? Some type of consensus? Just to understand the subjectivity/objectivity of this phase.

- Page 3 - Robo2VLM begin —> begins

**Dataset Code Accessibility:**

Yes

**Dataset Code Comments:**

The provided code, documentation, and data are clearly organized and accessible.

**Ethical Comments:**

The robot data is compiled from public datasets, which are properly cited. The generated VQA data has been manually reviewed and presents no significant ethical concerns.

**Ethical Considerations:**

No, there are no or only very minor ethics concerns

**Final Justification:**

The rebuttal presented by the authors brings interesting improvements to the evaluation of the work, together with clarifications to the raised concerns. Authors will also include results of the model trained on additional public bechmarks, for reference, in their final version. Considering all this improvements/additions, which strengthen the significance of the presented contribution, I'm raising my final rating.

**Limitations Weaknesses:**

The main weakness I find is regarding the evaluation of the presented work. I miss a bit more thorough experimentation that could help clarify or answer the following points, and therefore strengthen the demonstration of the impact of the presented framework and data:

- Why only multiple-choice questions? Recent works start to propose open-question benchmarks which are more challenging and often more realistic use cases.
- Regarding the strategies to mitigate and study the “impact” of multiple-choice questions, why not include a “blind” baseline that only takes the question and options (no visual info)? That will show how much the LLMs can “guess” from pure contexts or discarded options and how much improvement comes from visual grounding, as shown in some recent works [1]

- The presented benchmark is built from processing data compiled from several existing datasets, but it is not clearly discussed how the different sources may introduce biases in the system or results. For example, it would be interesting to see the results aggregated by source?

- To demonstrate further impact and significance, it would be interesting to know if training on the proposed data actually improves the VLM in other benchmarks that require the capabilities that this work targets to improve? (Reasoning about spatial or interaction elements), such as the benchmarks used in [1]

[1] “K. Ranasinghe, et al. Understanding long videos in one multi-modal language model pass. ICLR, 2025.”


There are also a few unclear points that could be better explained or discussed to better define the contribution and significance of the work:

- The current system reduces the temporal information from the recordings to single-image questions (given a keyframe, answer the question?). Isn’t this strategy losing somehow the temporal consistency implicit in the robot sequential recordings? Is the temporal information used somehow? Or just with the trajectory visualization superposed in the image as shown in some example figures in the paper?

 - Why are all the benchmark questions  guided by the pre-defined templates and not let the system generate more open questions? Isn’t this introducing unnecessary biases in the data?

**Strengths Contributions:**

- The work presented is timely and well motivated on some of the current weaknesses in VLMs, regarding more precise spatial and planning understanding.
- The main contribution of this work is a novel data generation framework, robo2vlm, to build challenging VQA data (requiring spatial and planning reasoning) from multi-modal robot trajectory recordings.
- An interesting contribution in this work is the exploration of a novel paradigm to use real robot recordings to generate data that can help improve existing VLMs. This makes robo2vlm-1 a unique and original dataset in this regard.
- Clear organization and description of the work presented in the main paper, including the key ideas and components of the strategy for question-answer generation from the robot recordings, and the content of the dataset released. Supplementary material covers well in detail all the steps of the VQA data generation framework and interesting analysis and statistics of the benchmark built as an example.
- It is notable and interesting the effort and strategies followed to include distractors and “None-of-the-above” alternatives to mitigate standard issues with multiple-choice tasks.
- The experimental validation illustrates first the challenge of the built benchmark for state-of-the-art open-source VLMs, and showcases how fine-tuning one of them (LlaVA) with the built dataset actually improves its performance on many of the challenging tasks posed in the presented dataset.
- Additionally, the provided code, documentation, and data are clearly organized and accessible.

---

> ### Author Rebuttal · Authors · 2025-07-31
>
> Thank you so much for the thoughtful and encouraging feedback\! We truly appreciate your recognition of the novelty and relevance of our work, as well as your detailed comments. We will address the limitations that you mentioned and clarify in the final version. Following are for the discussion of the limitations:
>
> * **Why only multiple-choice questions? Recent works start to propose open-question benchmarks which are more challenging and often more realistic use cases.**
>
> Thank you for the great suggestion\! We agree that using an open-ended format can be helpful.
>
> In this paper, we focus on multiple choice questions because it is a common standard in visual question answering. We carefully design distractors in each multiple choice question (we are glad that you like our distractor design\!). To design unambiguous distractors, we use procedural algorithms based on robotics domain knowledge and grounded in multimodal sensory data, including end-effector pose, gripper state, and force-torque readings. For example, distractor arrows in spatial prediction questions differ significantly in projected 2D angle; in depth perception questions, distractor points have depth values that differ meaningfully from ground truth. This enables precise calibration of model capabilities and reveals detailed failure modes (e.g., depth confusion, misaligned spatial attention). We will clarify this in the final version.
>
> We think using open questions is a very promising future direction. It is good for the model without extra instruction tuning and benefits failure diagnosis or human-in-the-loop correction (e.g., listing all objects in a video). However, for benchmarking and understanding model ability in the current Robo2VLM, using open questions is challenging to match to a semantically complex sentence such as the trajectory understanding and other goal-conditioned reasoning. In the future work, we will explore open-ended questions with human evaluation and LLM-as-a-judge.
>
> * **using “blind” VLM (not consider images, just use LLM) as baseline for comparison**
>
> Thank you for the great idea and introducing a new reference\! We will present comprehensive results and make proper citations to the blind VLM work. We conducted an initial ablation study on the “Blind” VLM setting by randomly sampling 1,000 VQA questions from the Robo2VLM-1 dataset, while keeping all other experimental settings consistent with Section 5.1. For the sampled subset, the average number of choices per question is 4.65, yielding a random baseline accuracy of approximately 21.5%. To evaluate the impact of visual input removal, we tested both GPT-4o (gpt-4o-2024-08-06), GPT-4o-mini (gpt-4o-mini-2024-07-18),  llava 1.5 \- 7B , and Qwen-7B under the “Blind” condition. Here are initial results of compare standard VLM with blind VLM in accuracy:
>
> | Model Name | Blind LLM | Standard VLM |
> | :---- | :---- | :---- |
> | GPT-4o-mini | 16.0% | 29.7% |
> | ChatGPT-4o | 21.7% | 41.5% |
> | llava 1.5 \- 7B | 22.4% | 22.6% |
> | Qwen-7B | 24.3% | 34.8% |
>
> The result suggests that the templated question does not provide hints to a specific outcome, so the blind model performance is close to random guess. This can also be reflected by the supplementary materials’s choice distribution (Section G). We will add complete results in the final version.
>
> * **The presented benchmark is built from processing data compiled from several existing datasets, but it is not clearly discussed how the different sources may introduce biases in the system or results. For example, it would be interesting to see the results aggregated by source?**
>
> Thank you for raising this important point. Indeed, the Open X-Embodiment (OXE) dataset is collected by 21 institutions worldwide. While OXE aims to be broad and inclusive, different scene setups, robot embodiment, teleoperator and task complexities may introduce biases. We will acknowledge the bias explicitly in the limitation section.
>
> We compiled the results shown in paper Table 3 as in the table below. The table aggregates the accuracy based on the data collection institution and the robot embodiment used by the institution. The results suggest that different data sources have slight variance due to visual or task complexities, but good overall model performance can generalize to a variety of sources. These results seem encouraging:  performance on Robo2VLM from diverse data sources may reflect generalization.
>
> | Model  / institution  | asu | austin | berkeley | droid | google  | nyu | stanford | Freiburg | Austin  |
> | ----- | ----- | ----- | ----- | :---- | ----- | ----- | ----- | ----- | :---- |
> | **embodiment** | **UR5** | **Franka** | **UR5** | **Franka** | **Google Robot** | **Hello Stretch** | **Kuka** | **Franka** | **Franka** |
> | **Llava-v1.6** | 32.7% | 24.2% | 25.6% | 21.7% | 26.2% | 16.7% | 34.3% | 23.0% | 26.9% |
> | **Qwen2.5-VL-7B** | 37.1% | 30.5% | 35.1% | 24.8% | 39.1% | **33.3%** | 30.3% | 36.4% | 32.5% |
> | **Qwen2.5-VL-32B** | **40.0%** | **38.3%** | **42.6%** | **29.2%** | **50.7%** | **33.3%** | **51.2%** | **41.2%** | **37.6%** |
> |  |  |  |  |  |  |  |  |  |  |
>
> We will incorporate the results associated with the insights in the final draft.
>
> * **Evaluate on other benchmarks**
>
> This is a very insightful suggestion\! Evaluating on more benchmarks is indeed a good way to show the Robo2VLM model’s generalization capability. Due to the interest of time frame and the scope of the paper, we plan to incorporate the results of the fine-tuned model on other benchmarks in the final version.
>
> * **Why are all the benchmark questions guided by the pre-defined templates and not let the system generate more open questions? Isn’t this introducing unnecessary biases in the data?**
>
> This is a good point in highlighting the contribution of the paper. In Robo2VLM, all questions and correct answers and distractors are grounded by robotics domain knowledge [1] and real multimodal sensory data. Therefore, we acknowledge this is a tradeoff between question scope and correctness. We will clarify that in the final draft.
>
> * **Single frame vs temporal consistency**
>
> Thanks for the important observation. We structure the robot trajectories into different manipulation phases following the robotics community standard [2].  Within the phase, the VQA can be formulated on short-horizon unitary actions (such as approaching the target, contact, moving away). Frames within one manipulation phase have minimal temporal dependency, so use a single frame. We will make clarification in the final version.
>
> [1]  Henrik I. Christensen and Gregory D. Hager. Sensing and estimation. In Bruno Siciliano and Oussama Khatib, editors, Springer Handbook of Robotics, Springer Handbooks, pages 91–112. Springer, 2016
> [2] Makoto Kaneko. Manipulation and Interfaces. Springer Handbook of Robotics, Springer Handbooks, pages 893–1126. Springer, 2016

---

> > ### Comment · Reviewer_FBtn · 2025-08-05
> > **rebuttal follow up**
> >
> > Thanks for the detailed rebuttal. I appreciate the effort on clarifying the raised concerns and including new information. The clarifications and additional experiments included (blind VLM and results grouped by source type) bring interesting insights to the experimental analysis, and the clarifications provided address well my main concerns. I agree multiple choice questions can be out of the scope of the present paper.
> >
> > I'm curious if the authors already have some prelimimary results to include from other benchmarks (as they mention that they will include results from the fine-tuned model on other benchmarks in the final version), I mean to see them before the discussion period finishes.
> >
> > With this and the previously mentioned improvements to the evaluation work, together with the clarifications discussed in the rebuttal, I find the significance of the presented contribution more strongly supported and I'm happy to raise my final rating.

---

### Note · Authors · 2025-08-14

We would like to thank the extensive positive and constructive feedback from all the reviewers! We appreciate reviewers acknowledging our novel data generation paradigm, our procedural algorithm based on robotics domain knowledge, and the overall paper presentation. We would also like to thank reviewers for pointing out the multiple choice format design consideration. Through the rebuttal and discussion period, we believe that we reached a consensus that using multiple choice can facilitate unambiguous evaluation and distractor design. With the helpful suggestions from the reviewers, we supplemented additional evaluation results on blind VLM, closed source VLM (GPT-4o), and evaluation by different data collection sources.  We acknowledge that some evaluation results might be left for future work due to the scope of the paper. We will reflect all the points mentioned by the rebuttal and modify the final paper based on the reviews.

---

### Decision · Program_Chairs · 2025-09-18

**Decision:**

Accept (spotlight)

**Comment:**

The paper proposes ROBO2VLM, a VLM-based framework to generate visual question answering (VQA) data from tele-operated robot trajectories consisting of a sequence of observations (e.g. RGB images, gripper feedback, etc).  The framework is applied to 176K trajectories from Open X-Embodiment, to create a dataset of 684K questions covering a range of different robotic manipulation tasks.  The questions are designed to assess various reasoning abilities required for manipulation (e.g. reasoning about spatial relations, task goals, etc.) The main contributions of the work are 1) the proposed framework, 2) the generated dataset, and 3) a set of experiments comparing the performance of different VLM models on the dataset.

All four reviewers were positive on the work and advocates for acceptance.  Reviewers found the dataset and framework to be valuable, indicating the paper to be well organized, with clear descriptions for both the key ideas and the details for the data generation.  Reviewers also indicated that they found the work to be timely and well motivated [FBtn], and the question prototypes to be well-designed [fh7u].

Reviewers initially expressed concerns that the experiments were somewhat limited [FBtn, Bctk, r1zP] had some questions regarding the design and limitations of the framework and resulting dataset [FBtn, fh7u, r1zP].

The AC agrees with the reviewers that the framework + dataset would be valuable to the community, and that the work is well executed.  The AC The AC recommends acceptance and encourages to improve the paper for the camera ready based on feedback from the reviewers, including integrating additional experiments provided during the rebuttal, and clarifications and discussions.